# Fbxo4-mediated degradation of Fxr1 suppresses tumorigenesis in head and neck squamous cell carcinoma

Shuo Qie[1], Mrinmoyee Majumder[1,2], Katarzyna Mackiewicz[1], Breege V. Howley[1], Yuri K. Peterson[3], Philip H. Howe [1], Viswanathan Palanisamy[1,2] & J. Alan Diehl[1]

The *Fbxo4* tumour suppressor is a component of an Skp1-Cul1-F-box E3 ligase for which two substrates are known. Here we show purification of SCF^Fbxo4 complexes results in the identification of fragile X protein family (FMRP, Fxr1 and Fxr2) as binding partners. Biochemical and functional analyses reveal that Fxr1 is a direct substrate of SCF^Fbxo4. Consistent with a substrate relationship, Fxr1 is overexpressed in Fbxo4 knockout cells, tissues and in human cancer cells, harbouring inactivating *Fbxo4* mutations. Critically, in head and neck squamous cell carcinoma, Fxr1 overexpression correlates with reduced Fbxo4 levels in the absence of mutations or loss of mRNA, suggesting the potential for feedback regulation. Direct analysis reveals that *Fbxo4* translation is attenuated by Fxr1, indicating the existence of a feedback loop that contributes to Fxr1 overexpression and the loss of Fbxo4. Ultimately, the consequence of Fxr1 overexpression is the bypass of senescence and neoplastic progression.

---

[1] Department of Biochemistry and Molecular Biology, Medical University of South Carolina, Charleston, SC 29425, USA. [2] Department of Oral Health Sciences and Centre for Oral Health Research, College of Dental Medicine, Medical University of South Carolina, Charleston, SC 29425, USA. [3] Department of Drug Discovery and Biomedical Sciences, Medical University of South Carolina, Charleston, SC 29425, USA. Correspondence and requests for materials should be addressed to J.A.D. (email: diehl@musc.edu)

Protein ubiquitylation controls protein stability, endocytosis, trafficking, DNA damage repair and cell signalling depending on the lysine residue (K6, K11, K27, K29, K33, K48 and K63) within the ubiquitin molecules that is modified[1]. K48-linked ubiquitylation governs proteasome-mediated protein degradation, through which it controls gene transcription, cell cycle progression, cell proliferation/growth as well as cell survival[2]. Fbxo4 belongs to the F-box protein family, defined by an F-box motif first noted in cyclin F[3, 4]. F-box proteins serve as the substrate recruitment factors for the SCF (S-phase kinase-associated protein 1 (Skp1)-Cullin 1-F-box) E3 ligases.

The disruption of the balance between protein translation and degradation directly contributes to cell transformation, tumorigenesis and tumour progression[3]. Fbxo4 is a tumour suppressor, and its tumour suppressing activity has been linked to the dysregulation of cyclin D1 proteolysis[5]. Fbxo4 missense mutations occur with a frequency of ~14% in human oesophageal squamous cell carcinoma (ESCC) and 10% in melanoma, accounting for cyclin D1 accumulation and tumorigenesis[3, 4].

Two SCF[Fbxo4] substrates have been identified: cyclin D1 and telomeric-repeat factor 1 (TRF1)/Pin2[6, 7]. Fbxo4 recognises cyclin D1 following glycogen synthase kinase 3β (GSK3β)-mediated Thr-286 phosphorylation[6, 8]. Fbxo4 is also activated by GSK3β via

**Fig. 1** Fbxo4 directly interacts with Fxr1. **a** Co-immunoprecipitation of Fbxo4 and Fxr1; arrows indicate Fbxo4 bands. **b** Endogenous Fbxo4 co-immunoprecipitates with Fxr1. **c** Ribbon diagram of the Fbxo4:Trf1 heterodimer, PDB:3L82. Fbxo4 is coloured in grey and Trf1 is in purple. **d** Intermolecular interactions between Fbxo4 and Trf1 derived from the PDB:3L82. **e** Ribbon diagram of the predicted interaction of Fbxo4 and an Fxr1 homology model. Fbxo4 is coloured in grey and Fxr1 is in purple. **f** Intermolecular interactions between Fbxo4 and Fxr1. HB is hydrogen bond, HYD is hydrophobic interaction, and ION an ionic bond. **g** Alignment of a semi-conserved motif in Trf1 and Fxr1. Identical amino acids are highlighted. Residues forming intermolecular bonds in Trf1 are boxed in blue, while residues mutated in this work are boxed in turquoise and magenta. Identity in this region was 30%, while similarity was 65%. **h** Fbxo4 E379A and E380A mutations disrupt the interaction between Fbxo4 and Fxr1. **i** Fbxo4 I377M mutation also disrupts the interaction between Fbxo4 and Fxr1. **j** Fxr1 V178A suppresses, while L189A Fxr1 enhances the interaction between Fbxo4 and Fxr1

phosphorylation, which is necessary for its dimerisation and E3 ligase activity[8]. Ubiquitylation of TRF1/Pin2 regulates telomere lengthening[9, 10] and in contrast to cyclin D1, Fbxo4 recognition is not dependent upon TRF1/Pin2 phosphorylation.

To identify Fbxo4 substrates, liquid chromatography-tandem mass spectrometry (LC-MS/MS) was utilised to analyse the Fbxo4 co-purifying proteins. Fragile X mental retardation syndrome proteins family (FMRP, Fxr1 and Fxr2) were identified as putative substrates. Notably, *Fxr1* is overexpressed in several cancers and its expression correlates with poor prognosis in patients with lung squamous cell carcinoma, as well as non-small cell lung cancer, ovarian cancer, breast cancer, and head and neck squamous cell carcinoma (HNSCC)[11, 12]. Herein, we demonstrate that SCF$^{Fbxo4}$ ubiquitylates and targets Fxr1 for proteasome degradation. Conversely, overexpression of *Fxr1* facilitates the bypass of senescence and tumour progression.

## Results

**Fxr1 is a Fbxo4 interacting protein**. To identify substrates of the SCF$^{Fbxo4}$ E3 ligase, *Fbxo4*−/− MEFs reconstituted with Flag-Fbxo4 or Flag-Fbxo4ΔF, which binds to substrates without recruiting E1 or E2 enzymes[6, 7], were treated + /− MG-132 for 6 h, and subjected to immuno-affinity purification. Co-purified proteins were identified by LC-MS/MS (Supplementary Fig. 1a). Among the interactions detected, all three members of the FMR family (FMRP, Fxr1 and Fxr2) were identified. To validate binding, Flag-Fbxo4 and Flag-Fbxo4ΔF were co-expressed with myc-tagged Fxr1 in HEK293T cells; cyclin D1 was co-expressed as a positive control. Fxr1 was readily detectable in Fbxo4 precipitates (Supplementary Fig. 1b), and conversely, Fbxo4 co-precipitated with myc-Fxr1 (Fig. 1a). Fxr1 also interacted with components of the SCF complex, including Skp1, Cul1 and Rbx1 (Supplementary Fig. 1c). Endogenous Fxr1 and Fbxo4 also co-precipitated (Fig. 1b and Supplementary Fig. 1d). FMRP also co-precipitated with Fbxo4; however, this is likely mediated by heterodimerisation with Fxr1 (Supplementary Fig. 1e). Fbxo4-Fxr2 interactions could not be confirmed (Supplementary Fig. 1b). *Fxr1* was chosen for further investigation due to its cancer relevance.

In order to determine the potential interacting sites between Fbxo4 and Fxr1, a model of the Fbxo4:Trf1 heterodimer[7] was created using the Fbxo4 X-ray data and the X-ray data of *FMRP*, an *Fxr1* homologue. Twelve interactions were identified across a broad interface that implicates nine amino acid (aa) resides, forming five hydrogen bonds, six hydrophobic interactions and one ionic bond in Trf1 (Fig. 1c, d), in which two areas of interaction were identified: $A^{64}$–$E^{69}$ and $S^{104}$–$I^{123}$; four inter-molecular bonds were found in the latter region. Furthermore, a pairwise sequence alignment was performed to identify common regions or motifs between Fxr1 and Trf1, especially, in areas implicated in Fbxo4:Trf1 interaction. One area of high similarity was identified corresponding to $S^{104}$–$I^{123}$ in Trf1 and $A^{173}$–$I^{192}$ in Fxr1 (Supplementary Fig. 2a). Of particular interest was the existence of four hydrophobic interactions ($I^{109}$, $L^{115}$, $L^{120}$ and $I^{123}$ in Trf1) found in the Fbxo4:Trf1-interacting model, corresponding to $V^{178}$, $L^{184}$, $L^{189}$ and $I^{192}$ in Fxr1 (Fig. 1g).

To further investigate the possible interactions, a theoretical three-dimensional heterodimer model of Fbxo4 and Fxr1 was created using the published X-ray structure of Fbxo4, a homology model of *Fxr1*, and bimolecular docking using CluPro[13]. *FMRP*, the homology model of *Fxr1*, was used as a template, which has an overall similarity of 81% within the model (Supplementary Fig. 2b). The bimolecular docking predictions indicated that all the top consensus models used the same structural interacting

**Table 1 ClusPro bimolecular docking of Fbxo4 with Fxr1**

| Cluster[a] | Members | Weighted score | Model name | Model file |
|---|---|---|---|---|
| 1 | 243 | −1054.7 | C1_243 | C1_243.pdb |
| 2 | 90 | −874.4 | C2_90 | C2_90.pdb |
| 3 | 86 | −930.0 | C3_86 | C3_86.pdb |
| 4 | 60 | −928.7 | C4_60 | C4_60.pdb |
| 5 | 49 | −926.2 | C5_49 | C5_49.pdb |
| 6 | 48 | −923.7 | C6_48 | C6_48.pdb |
| 7 | 42 | −878.5 | C7_42 | C7_42.pdb |
| 8 | 40 | −839.6 | C8_40 | C8_40.pdb |
| 9 | 34 | −839.6 | C9_34 | C9_34.pdb |
| 10 | 29 | −866.9 | C10_29 | C10_29.pdb |

[a]The top 10 clusters using the balanced model are shown. Members indicate the number of similar poses found in each cluster (RMSD ≤ 10 Å). Weighted score represents the lowest energy for the given cluster. Model names were created by combining the cluster number with the number of members in that cluster. Model file names refer to the Supplementary Data available for download

interface between Fbxo4 and Trf1. There was more uncertainty in the Fxr1 interface, however, 60% of the top 10 models used the same interface but with slightly different rotations (Table 1, Supplementary Fig. 3 and Supplementary Data 1–10). The predicted model has five of the nine corresponding residues from Fbxo4:Trf1-interacting model. Of the 11 residues in Fxr1 that were predicted to interact with Fbxo4, the aa175-aa180 region was identical in the region implicated in the sequence alignment analysis (Fig. 1e–g). These data suggest the interaction of Fbxo4 with either Trf1 or Fxr1 uses, in part, a similar interface driven by the hydrophobic residues in C-terminal amphipathic helix.

**Biochemical identification of sites for their interaction**. According to the 3D docking model, the C-terminus of Fbxo4 should interact with Fxr1 (Fig. 1e, f). Biochemical screening was performed to identify sites that are important for their interaction. Of the Fbxo4 mutants evaluated (ΔN, ΔF, ΔC2 and ΔC3; Supplementary Fig. 4a), Fbxo4ΔN, ΔC2 and ΔC3 were defecting in binding (Supplementary Fig. 4b). Reduced binding by Fbxo4ΔN suggests that Fbxo4 dimerisation is needed for binding. According to the model structure, E379 and E380 within the ΔC3 region should make direct contact with Fxr1 (Fig. 1h). Alanine substitution at these residues disrupted Fbxo4 and Fxr1 binding, while the double mutation of C341W/A354R residues that mediate Trf1 interaction[14] failed to disrupt their binding (Fig. 1h, i and Supplementary Fig. 4c). Additionally, a cancer-derived Fbxo4 mutant, I377M[14], was assessed for Fxr1 binding. Fbxo4I377M, which does not bind cyclin D1, was defective in Fxr1 binding (Fig. 1i and Supplementary Fig. 4c), demonstrating the substrate-binding domain for Fxr1 and cyclin D1 is overlapping. Mutations that impair Fbxo4 phosphorylation[3] did not inhibit substrate binding (Supplementary Fig. 4d).

With regard to residues in Fxr1 that mediate the binding, alignment of Fxr1 with Trf1 revealed a highly conserved region (Fig. 1g). Mutational analysis revealed that a V178A mutation disrupted binding while, strikingly, a L189A mutation facilitated binding (Fig. 1j and Supplementary Fig. 4e), indicating the changing of local three-dimensional structure strongly modifies binding. Of note, V178R also enhanced their binding (Supplementary Fig. 4f).

The data presented above demonstrate that the region (aa173-aa192) in Fxr1 mediates Fbxo4 recognition and suggests a direct interaction. Critically, these residues are not conserved between Fxr1 and FMRP, suggesting FMRP co-purification with Fbxo4 might not reflect direct binding. Since our modelling data, and

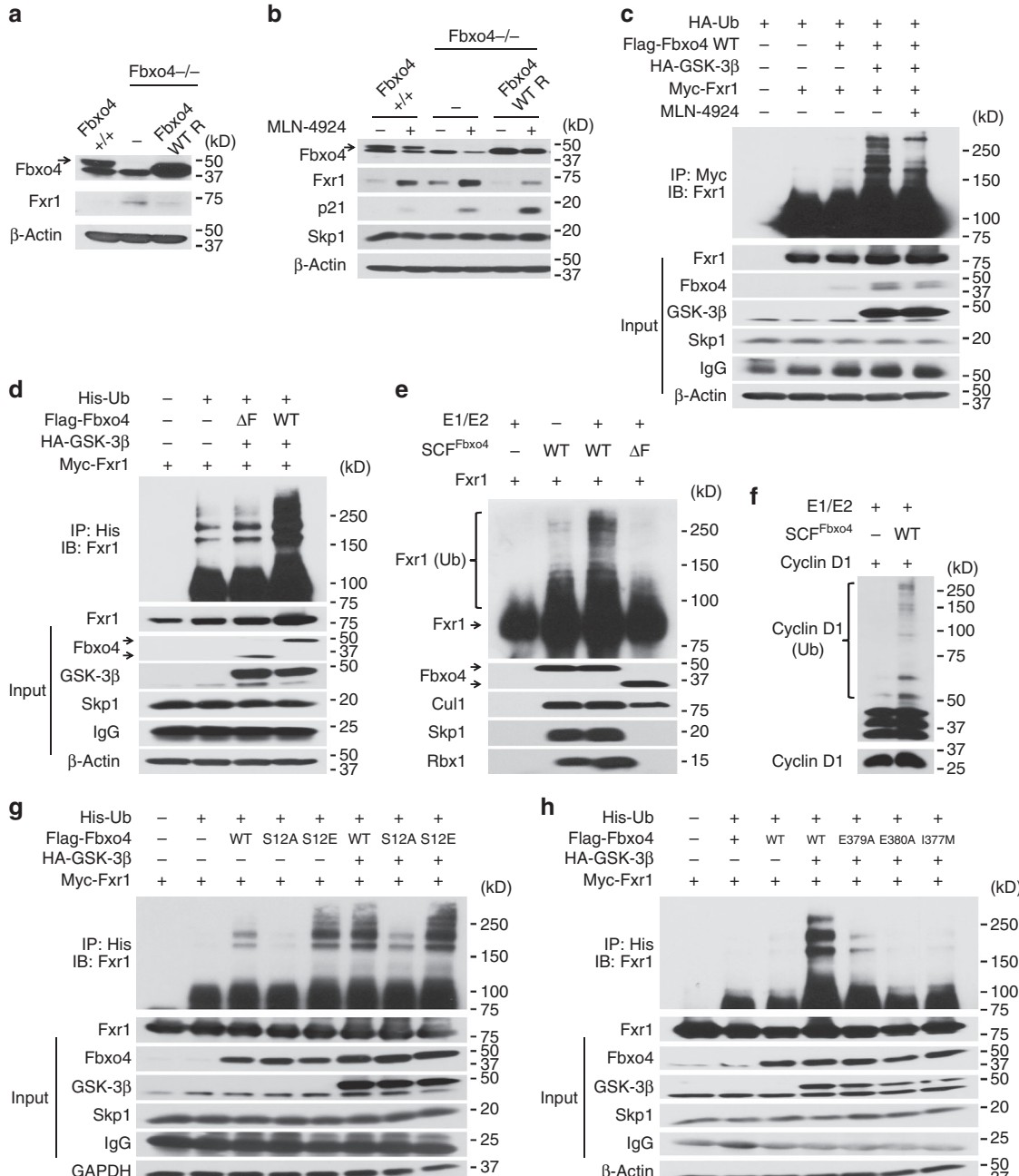

**Fig. 2** Fbxo4 ubiquitylates Fxr1 both in vivo and in vitro. **a** Fbxo4 reconstitution reduces Fxr1 levels in *Fbxo4−/−* MEFs. Note, arrow indicates Fbxo4; MEFs harbour a non-specific band that migrates below bona fide Fbxo4. **b** MLN-4924 rescues Fxr1 levels in both *Fbxo4 + / +* and −/− MEFs. **c** Fbxo4-induced Fxr1 ubiquitylation is enhanced by GSK3β co-expression and suppressed by MLN-4924 treatment for 6 h. **d** WT but not ΔF Fbxo4 ubiquitylates Fxr1 in vivo. **e** In vitro assay illustrates Fxr1 is ubiquitylated by Fbxo4. **f** Cyclin D1 ubiquitylation is used as a control for in vitro assays. **g** WT and S12E Fbxo4 enhance ubiquitylation of Fxr1 in vivo. **h** E379A, E380A and I377M Fbxo4 mutants lose the ability to ubiquitylate Fxr1 in vivo

in vitro ubiquitylation assays reveal Fxr1 to be a direct substrate, combined with published data that Fxr1 and FMRP heterodimerise, we considered the possibility that Fbxo4-FMRP co-precipitation is mediated by Fxr1. To address this, we assessed Fbxo4-FMRP co-precipitation in *Fxr1* knockdown cells. Indeed, *Fxr1* knockdown reduced FMRP co-precipitation (Supplementary Fig. 1e). These data reveal that Fbxo4 binds uniquely and specifically to Fxr1.

**Fbxo4 regulates Fxr1 degradation in normal cells**. To determine whether Fbxo4 regulates Fxr1 accumulation, Fxr1 levels were

assessed in wild-type (WT) vs. *Fbxo4−/−* MEFs. Fxr1 levels were elevated in *Fbxo4−/−* MEFs relative to WT counterparts (Fig. 2a, b and Supplementary Fig. 5a); moreover, overexpression of WT *Fbxo4* reduced Fxr1 expression to near normal levels (Fig. 2a, b). Treatment with MLN-4924, an inhibitor of SCF activity through suppression of cullin neddylation[15], increased Fxr1 and p21$^{Cip1}$ levels, which was assessed as a control for MLN-4924 efficacy, suggesting ubiquitin-dependent degradation regulates steady state Fxr1 accumulation (Fig. 2b and Supplementary Fig. 5a). Quantitative reverse-transcription-polymerase chain reaction (qRT-PCR) demonstrated that *Fxr1* mRNA was marginally elevated (Supplementary Fig. 5b).

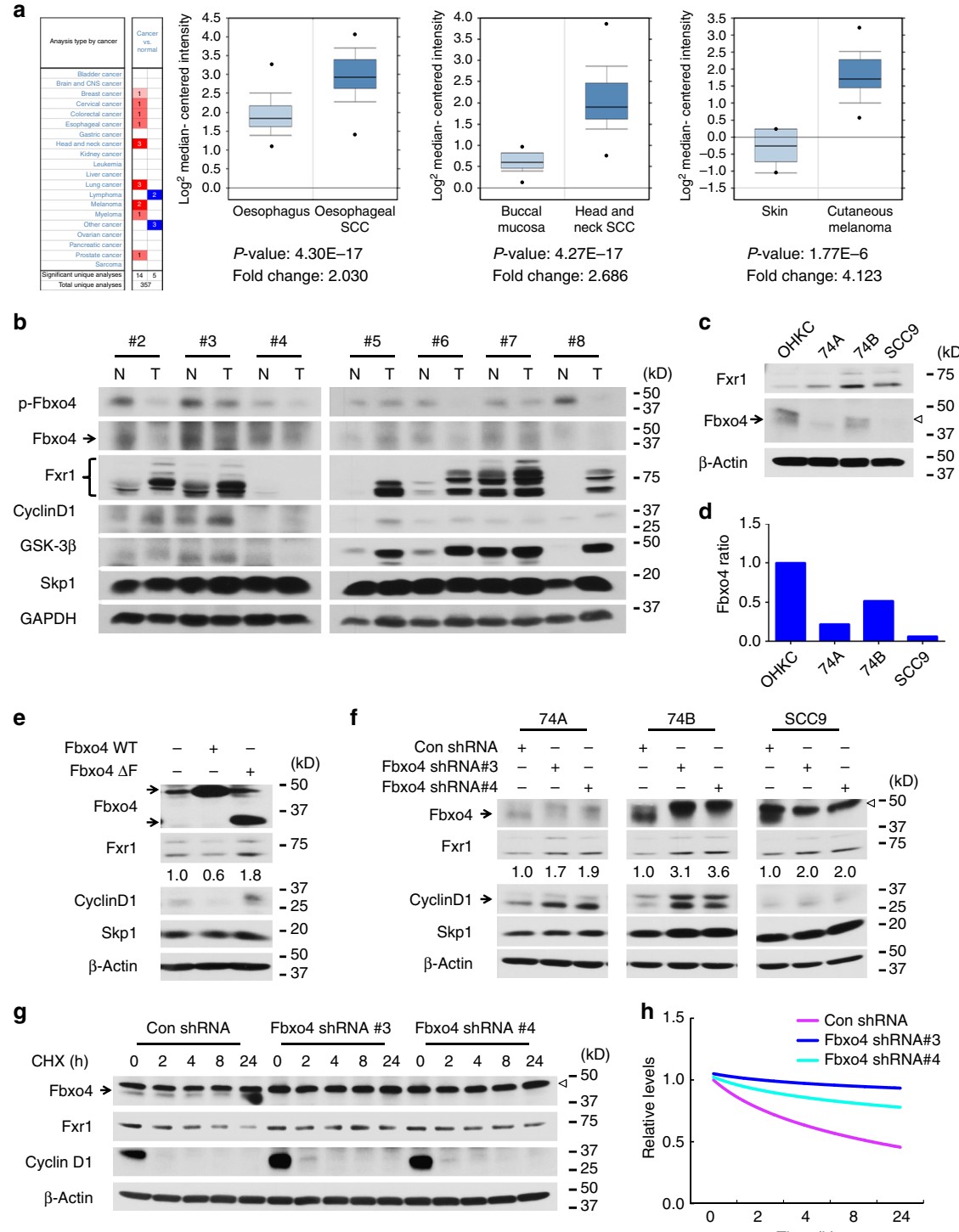

**Fig. 3** Genetic manipulation of *Fbxo4* alters Fxr1 expression in HNSCC cells. **a**, **b** Oncomine analysis reveals elevated *Fxr1* in human cancers (**a**) and a reverse correlation of Fbxo4 expression with Fxr1 in human HNSCC tissues and normal counterparts (**b**). All the data represent mean ± s.d. and were analysed by Student's *t* test. Oncomine Box-and-Whisker plots: median values are shown as horizontal bars; the upper and lower part of the box show the 75th percentile and the 25th percentile, respectively; the upper and lower part of the bar show the 90th percentile and the 10th percentile, respectively; the points show outlier values. **c** Comparison of Fxr1 and Fbxo4 levels in OHKC and human HNSCC cells. Empty triangle indicates non-specific band observed in some human cell lines. **d** Quantification of Fbxo4 bands shown in **c**, normalised to β-Actin. **e** WT but not Fbxo4ΔF suppresses Fxr1 expression in SCC9 cells. The numbers below Fxr1 bands indicate the band quantification. **f** *Fbxo4* knockdown triggers increased Fxr1 in HNSCC cells. Arrow indicates Fbxo4; empty triangle indicates nonspecific band. The numbers below Fxr1 bands indicate the band quantification. **g** Cycloheximide chase of Fxr1 levels in 74B cells following Fbxo4 knockdown. Arrow indicates Fbxo4; empty triangle indicates nonspecific band. **h** Quantification of Fxr1 turnover from **g**

**Fxr1 is an SCF^Fbxo4 substrate**. To address whether Fbxo4 catalyses ubiquitylation of Fxr1, HEK293T cells were co-transfected with myc-Fxr1, Flag-Fbxo4, ubiquitin with or without HA-GSK3β, and then denaturing immunoprecipitation was performed. WT Fbxo4 modestly increased Fxr1 ubiquitylation in the absence of ectopic *GSK3β*, while co-expression of GSK3β

dramatically increased SCF^Fbxo4-dependent Fxr1 ubiquitylation (Fig. 2c and Supplementary Fig. 5c, d). Polyubiquitylation was dependent upon K48 linkage, consistent with ubiquitin-mediated degradation (Supplementary Fig. 5e). GSK3β-dependent phosphorylation is necessary for Fbxo4 enzymatic activity (Supplementary Fig. 5f)[3]. GSK3β has also been implicated in promoting

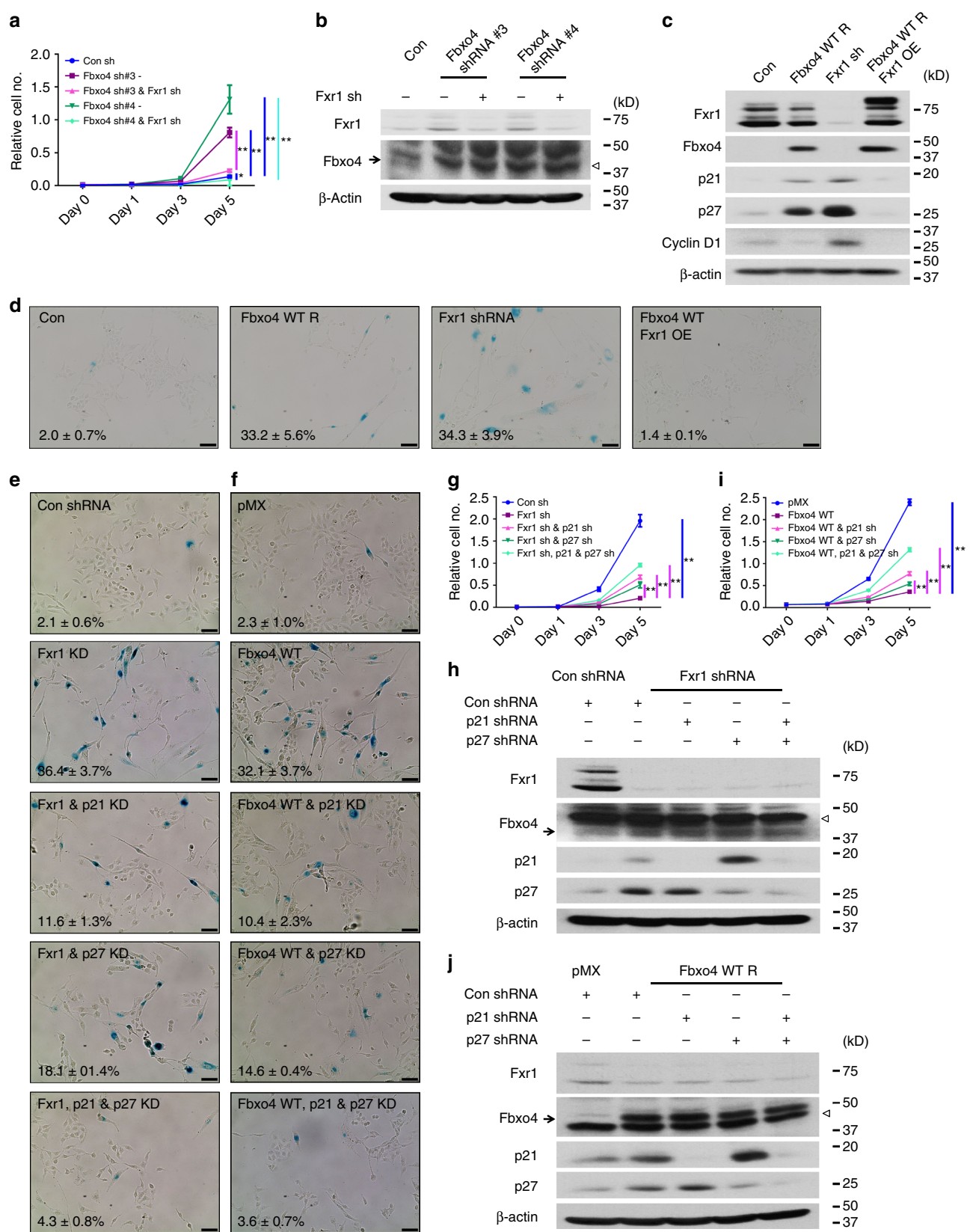

Fxr1 degradation[16]. Consistently, overexpression of *GSK3β* dramatically increased SCF$^{Fbxo4}$–dependent Fxr1 ubiquitylation (Fig. 2c and Supplementary Fig. 5c, d); while inclusion of MLN-4924 dramatically inhibited Fxr1 ubiquitylation. As an additional control, dominant negative Fbxo4ΔF was used; SCF$^{Fbxo4ΔF}$ did not increase Fxr1 ubiquitylation above background (Fig. 2d). SCF$^{Fbxo4}$ failed to catalyse ubiquitylation of FMRP (Supplementary Fig. 5g), consistent with evidence demonstrating lack of direct binding (Supplementary Fig. 1e)[17, 18].

While the data from cells strongly support a model where Fxr1 is a direct SCF$^{Fbxo4}$ substrate, it does not exclude the potential for an as yet unidentified component that mediates Fxr1 recognition. Therefore, we purified recombinant SCF$^{Fbxo4}$ generated in Sf9 cells. Purified SCF$^{Fbxo4}$ catalysed both Fxr1 and cyclin D1 ubiquitylation in vitro, while purified SCF$^{Fbxo4ΔF}$ was catalytically deficient (Fig. 2e, f and Supplementary Fig. 6a). Expression of phospho-mimetic Fbxo4S12E increased Fxr1 ubiquitylation, while an S12A mutant decreased polyubiquitylation (Fig. 2g and Supplementary Fig. 6b, c), demonstrating a role of Fbxo4 phosphorylation in Fxr1 ubiquitylation. Fbxo4 E379A, E380A and I377M mutants were also assessed for ubiquitylation of Fxr1 to ensure the direct interaction was required. Fxr1 ubiquitylation was not supported by these mutants both in vitro and in vivo (Fig. 2h and Supplementary Fig. 6d–f). Taken together, the biochemical data support SCF$^{Fbxo4}$ can directly ubiquitylate and degrade Fxr1 in a manner that depends upon GSK3β-mediated Fbxo4 phosphorylation.

**Fbxo4 regulates Fxr1 accumulation in HNSCC cells.** While inactivation of FMRP family proteins contributes to fragile X syndrome and mental retardation, mining of the data deposited in Oncomine specifically revealed elevated *Fxr1* in human cancers (Fig. 3a); in addition, western analysis revealed elevated Fxr1 protein in tumour relative to matched normal tissues in six out of seven samples (Fig. 3b). Approximately 30% of HNSCC exhibit DNA copy number alterations, which overlap with the *Fxr1* locus. Whether this is the sole contributor to Fxr1 overexpression has not been determined. Since Fbxo4 is inactivated in oesophageal squamous cancers[3], we considered the possibility that reduced Fbxo4 might also contribute to Fxr1 overexpression. We initially examined available cell lines established from oral cancers and noted that Fbxo4 expression is reduced in all three oral cancer cell lines UM-SCC-74A (74 A), UM-SCC-74B (74B) and SCC9 cells compared to normal human oral karotinocyte (OHKC); more importantly, reduced Fbxo4 correlated with increased Fxr1 levels (Fig. 3c, d). Consistent with reduced Fbxo4-dependent regulation in these cells, overexpression of WT *Fbxo4* reduced Fxr1, while Fbxo4ΔF was ineffective and actually increased Fxr1 (Fig. 3e and Supplementary Fig. 7a, b). Knockdown of *Fbxo4* with two independent shRNAs increased Fxr1 protein levels (Fig. 3f) and this corresponded with decreased protein degradation (Fig. 3g; quantification, 3 h); moreover, overexpression of WT *Fbxo4* instead of *Fbxo4ΔF* shortened the half-life of Fxr1 in HEK293T cells (Supplementary Fig. 7c). Ectopic *Fbxo4*

expression successfully antagonised Fbxo4 knockdown-mediated Fxr1 upregulation (Supplementary Fig. 7d). To demonstrate the role of GSK3β in regulating Fbxo4-mediated Fxr1 degradation, SB-216763 was utilised to treat HNSCC cells with or without ectopic *Fbxo4* expression. Consistent with ubiquitylation assay, inhibition of GSK3β kinase activity rescues Fxr1 downregulation-mediated by Fbxo4 (Supplementary Fig. 7e). To further corroborate the ubiquitylation findings, S12A, S12E, E379A, and I377M Fbxo4 were expressed in HNSCC cells. Only WT and S12E Fbxo4 but not inactive mutants effectively suppressed Fxr1 expression (Supplementary Fig. 7f–h), confirming ubiquitylation is crucial for Fxr1 downregulation in HNSCC cells.

**Biological function of Fbxo4–Fxr1 axis.** We next sought to address the role of Fbxo4-dependent regulation of Fxr1 in HNSCC cells. Initially, we reduced Fbxo4 levels in HNSCC cells using two distinct shRNA constructs. *Fbxo4* knockdown increased Fxr1 levels (Fig. 4b and Supplementary Fig. 8b, d) and promoted cell proliferation (Fig. 4a and Supplementary Fig. 8a, c, i–k). Consistent with elevated Fxr1 contributing to increased proliferation, concurrent Fbxo4 and Fxr1 knockdown inhibited cell growth in all three HNSCC cell lines (Fig. 4a, b and Supplementary Figs. 8a–d and 9a–d). Fxr1 controls cell division and senescence through regulation of *p21$^{Cip1}$* mRNA degradation[12, 19]. Fxr1 loss also reduces cell growth in soft agar assay (Supplementary Fig. 9e, f). The collective impact of Fxr1 loss is p21$^{Cip1}$ overexpression and cell senescence[12, 19]. Senescence is a state of permanent cell proliferation arrest[20, 21]. In contrast, Fxr1 overexpression facilitates senescence bypass and neoplastic growth in HNSCC cells[12]. We therefore reasoned that restoration of Fbxo4 levels would reduce Fxr1 and trigger senescence. Consistently, Fbxo4 expression increased SA-β-Gal staining to a similar degree as *Fxr1* knockdown (Fig. 4c, d and Supplementary Fig. 8e–h). Importantly, enforced Fxr1 expression concurrent with Fbxo4 overexpression resulted in senescence bypass consistent with Fxr1 being downstream of Fbxo4 (Fig. 4d and Supplementary Fig. 8g, h). *Fbxo4* overexpression or *Fxr1* knockdown resulted in increased expression of both p21$^{Cip1}$ and p27$^{Kip1}$ (Fig. 4c and Supplementary Fig. 8e, f). In contrast, coordinated *Fbxo4* and *Fxr1* overexpression prevented p21$^{Cip1}$ and p27$^{Kip1}$ induction, which is consistent with no induction of senescence.

To further illustrate the role of p21$^{Cip1}$ and p27$^{Kip1}$ in regulating cellular senescence and proliferation, shRNAs were utilised to knockdown one or both p21$^{Cip1}$ and p27$^{Kip1}$ upon *Fxr1* knockdown or *Fbxo4* overexpression (Fig. 4h, j). Double knockdown of both *p21$^{Cip1}$* and *p27$^{Kip1}$* antagonises cell senescence and proliferative suppression -induced by either *Fxr1* knockdown or *Fbxo4* overexpression; while for single knockdown, *p21$^{Cip1}$* knockdown provided a better rescue than *p27$^{Kip1}$* (Fig. 4e–g, i and Supplementary Fig. 9g, h), consistent with the compensatory upregulation of *p21$^{Cip1}$* by *p27$^{Kip1}$* knockdown (Fig. 4h, j). Fxr1-mediated *p21$^{Cip1}$* mRNA degradation has been clearly

**Fig. 4** Fxr1 promotes cell proliferation and inhibits senescence-induced by ectopic *Fbxo4* expression. **a** *Fxr1* knockdown reverses *Fbxo4* knockdown-induced cell proliferation of 74B cells. **b** Western blot shows Fbxo4 and/or Fxr1 Knockdown in 74B cells. Empty triangle indicates nonspecific band. **c** Expression of p21 and p27 in 74B cells upon *Fbxo4* overexpression, *Fxr1* knockdown and both *Fbxo4* and *Fxr1* overexpression. **d** β-Gal staining indicates senescent cells in 74B cells upon *Fbxo4* overexpression, *Fxr1* knockdown and both *Fbxo4* and *Fxr1* overexpression. The numbers show the percentage of β-Gal-positive cells in three independent experiments. **e, f** *p21* and *p27* knockdown rescue senescence in *Fxr1* knockdown (**e**) and *Fbxo4* overexpression (**f**) 74B cells. The numbers show the percentage of β-Gal-positive cells in three independent experiments. **g, i** *p21* and *p27* knockdown rescues cell proliferation in *Fxr1* knockdown (**g**) and *Fbxo4* overexpressing (**i**) 74B cells. **h, j** Western blots show the knockdown of *p21* and *p27* in *Fxr1* knockdown (**h**) and *Fbxo4* overexpressing (**j**) 74B cells. Empty triangle indicates nonspecific band. All the data represent mean ± s.d. and were analysed by Two-way ANOVA, followed by Fisher's LSD as post hoc test (*n* = 3). \**p* < 0.05; \*\**p* < 0.01. Scale bar, 10 μM

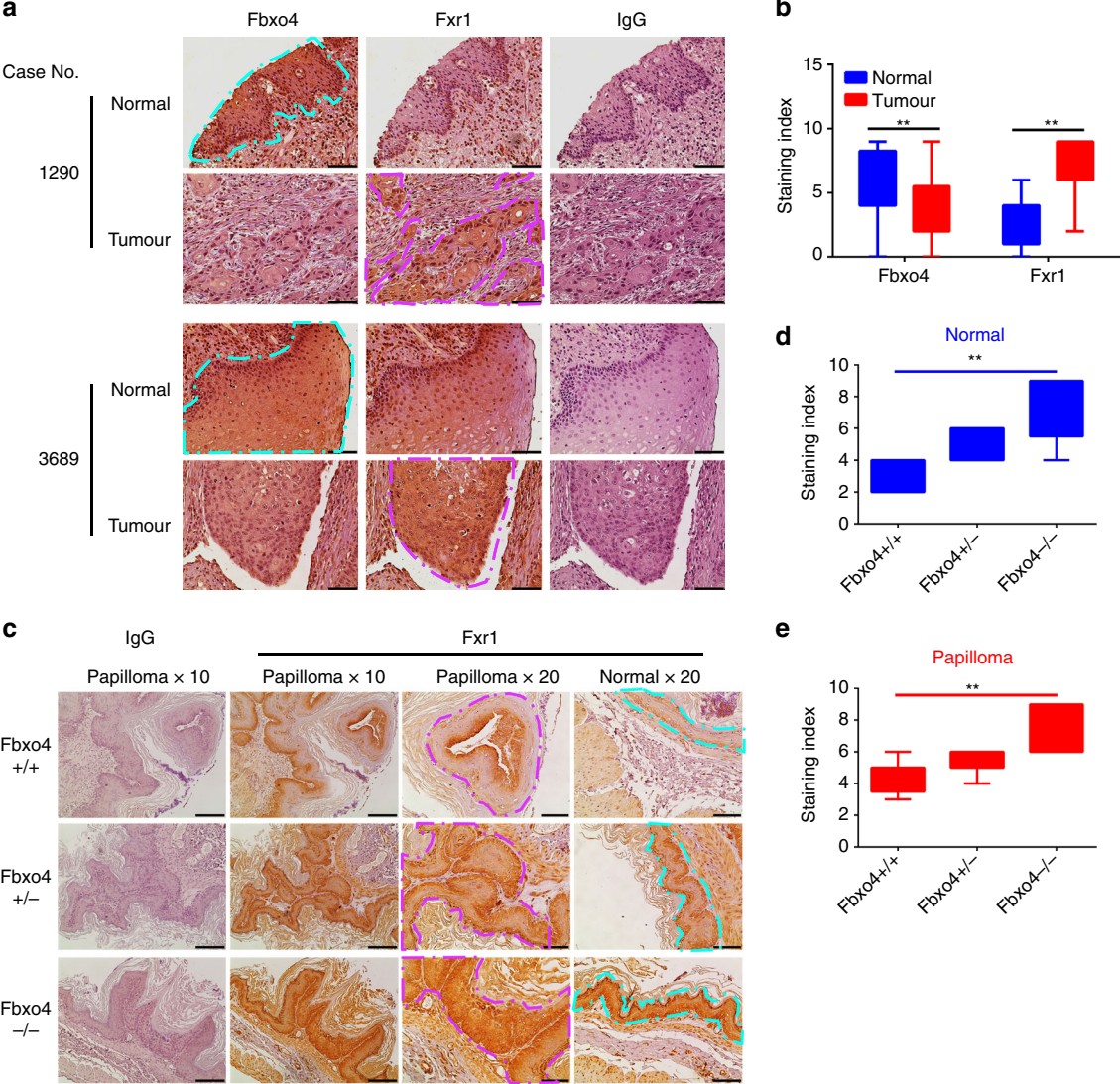

**Fig. 5** Fxr1 expression reversely correlates with Fbxo4 protein levels in both human HNSCC and mouse papilloma tissues. **a** Representative image of Fbxo4 and Fxr1 IHC staining in human HNSCC TMAs. **b** Quantification and statistical analyses of the stained TMA specimens. Analysis was performed using nonparametric Mann–Whitney $U$ test. **$p < 0.01$. **c** Representative illustration of Fxr1 IHC staining in papilloma-induced by NMBA in transgenic mice with $Fbxo4 + / +, + / −$ and $−/−$ genetic background. **d, e** Quantification and statistical analyses of the IHC stained normal (**d**) and papilloma (**e**) sections. Analyses were performed using nonparametric Kruskal–Wallis test. **$p < 0.01$. Box-and-Whisker plots: the upper and lower parts of the box show the 75th percentile and the 25th percentile, respectively; the bars outside of the box show the Min and Max values. Turquoise enclosed areas indicate normal epithelia; magenta-enclosed areas indicate either HNSCC or papilloma. Scale bar, 10 μM

demonstrated; *Fxr1* knockdown resulted in increased mRNA and protein levels of $p27^{Kip1}$, while no mRNA binding was detected by RNA-binding protein immunoprecipitation (RIP)[12], suggesting indirect regulation. Collectively, these data support a model wherein Fbxo4 downregulation results in Fxr1 overexpression and senescence bypass, allowing neoplastic growth.

**Fbxo4 loss and Fxr1 overexpression in HNSCC tumours.** Given the ability of Fbxo4 to antagonise Fxr1-dependent cell expansion in established cell lines, we next sought to assess the Fbxo4–Fxr1 relationship in clinical samples. Western analysis of lysates from frozen HNSCC tumour and adjacent normal tissues revealed Fxr1 elevation in tumour tissues; notably, Fbxo4 levels were reduced relative to normal in four out of six tumours with elevated Fxr1 (Fig. 3b). Although similar total Fbxo4 protein was presented in sample #7, low phospho-Ser12 Fbxo4 was observed (Fig. 3b), suggesting reduced SCF^Fbxo4 E3 ligase activity in this tumour. To

further interrogate the Fbxo4–Fxr1 regulatory axis in human head and neck cancer, serial sections of tissue microarrays (TMAs) with 36-paired normal and tumour tissue cores were used for immunohistochemical (IHC) staining and pathological assessment (Fig. 5a). IHC revealed reduced Fbxo4 levels specifically in malignant tissues, while Fxr1 staining inversely correlated with Fbxo4 levels (Fig. 5a, b). To corroborate the dysregulation of Fbxo4–Fxr1 axis, both ESCC and melanoma cells were utilised due to the presence of Fbxo4 mutations in these cancers[3, 14]. Fxr1 was elevated in TE10 cells, a cell line that harbours an S8R mutation that disrupts Fbxo4 dimerisation (Supplementary Fig. 10a)[22]. Moreover, melanoma cells with Fbxo4I377M mutation also exhibit increased Fxr1 levels (Supplementary Fig. 10b–d), consistent with Fbxo4-dependent regulation of Fxr1 in normal cells, and the disruption of this pathway in human cancers. Importantly, increased Fxr1 protein levels correlate more closely with *Fbxo4* loss and mutational status than with *Fxr1* mRNA accumulation (Supplementary Fig. 10b–d).

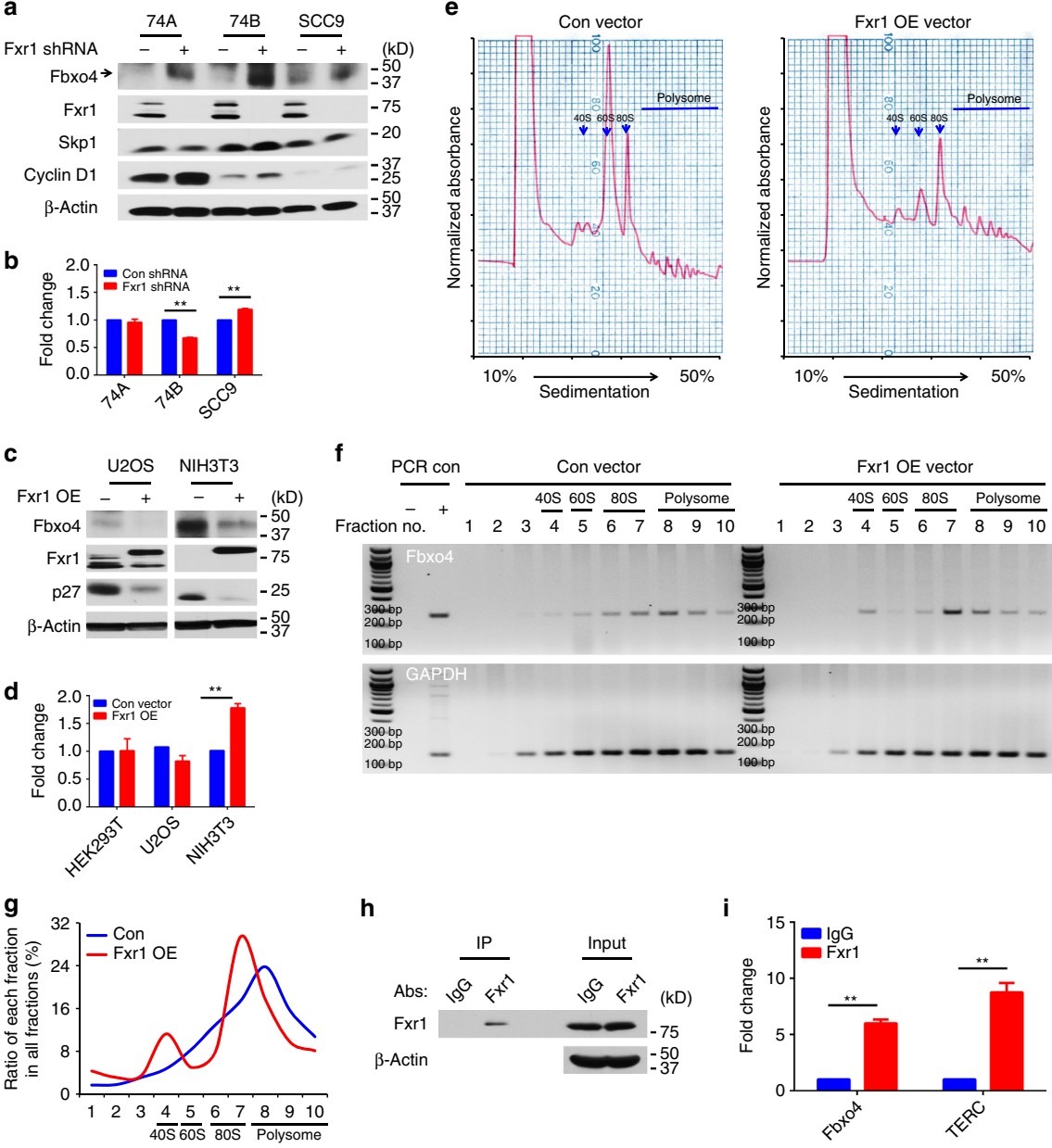

**Fig. 6** Negative feedback regulation of Fbxo4 by Fxr1. **a** Fbxo4 protein levels upon *Fxr1* knockdown in 74A, 74B and SCC9 cells. **b** *Fbxo4* mRNA levels upon *Fxr1* knockdown in 74A, 74B and SCC9 cells. **c** Fbxo4 protein levels upon ectopic *Fxr1* expression in U2OS and NIH3T3 cells. **d** *Fbxo4* mRNA levels upon ectopic *Fxr1* expression in HEK293T, U2OS and NIH3T3 cells. **e** Polysome profile in NIH3T3 cells with *Fxr1* overexpression. **f** DNA gel shows the RT-PCR products from serial polysome fractions. **g** Quantitative distribution of *Fbxo4* mRNA from (**f**). **h** Western blot from immunoprecipitation of Fxr1 in RIP analysis. **i** RIP analysis indicates Fxr1 interacts with *Fbxo4* mRNA. *TERC* mRNA is used as positive control. All the data represent mean ± s.d. and were analysed by Student's *t* test ($n = 3$). \*\*$p < 0.01$

To further assess Fbxo4–Fxr1 regulation in tumorigenesis, tumours that developed in *Fbxo4* +/+, +/− and −/− mice, treated with N-nitrosomethylbenzylamine (NMBA) to trigger SCC[23–25], were subjected to IHC staining. As reported[25], high papilloma incidence was observed in *Fbxo4* +/− (27/32, 84.4%) and −/− mice (20 out of 22, 90.9%) compared with +/+ mice (6 out of 21, 28.6%), $p < 0.01$ ($\chi^2$ test). Five papilloma plus the adjacent normal tissues were randomly selected for Fxr1 IHC staining from these cohorts. IHC revealed elevated Fxr1 in normal tissues from *Fbxo4*−/− and +/− mice relative to that in *Fbxo4* +/+ mice; an obvious elevation of Fxr1 was noted in papilloma of *Fbxo4* +/− and −/− mice compared to that in +/+ mice (Fig. 5c–e). These data demonstrate that Fbxo4 regulates

Fxr1 accumulation in vivo and that loss of Fbxo4 leads to Fxr1 overexpression in both normal and tumour tissues.

**Feedback regulation of Fbxo4 by Fxr1**. Although *Fbxo4* mutations have been found in human ESCC and melanoma[3, 14], additional mutations are only rarely observed in other human cancers (Supplementary Fig. 11a–g) (http://www.cbioportal.org/). However, we did note that following *Fxr1* knockdown, a corresponding increase of Fbxo4 protein levels was observed, suggesting a potential for an auto-regulatory feedback loop (Fig. 6a). Fxr1 regulates gene expression through direct interaction with mRNAs containing AU-rich elements (ARE), for example, *TNF-*

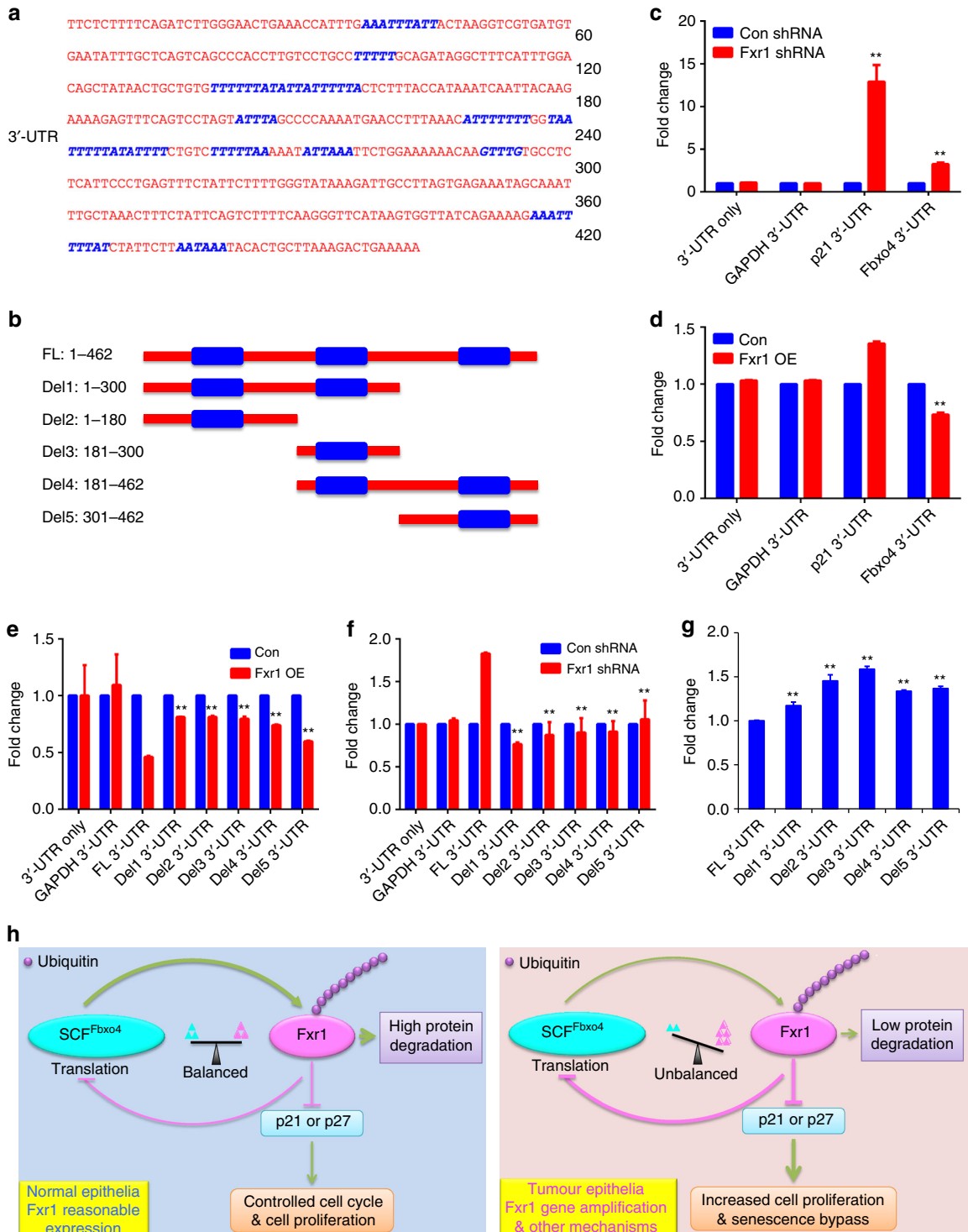

**Fig. 7** Identification of the AREs that control Fbxo4 translation by Fxr1. **a** Schematic illustration of AREs in 3′-UTR of *Fbxo4* mRNA; AREs are indicated in blue. ARE prediction is performed using AREsite2: http://rna.tbi.univie.ac.at/AREsite2/welcome. **b** Deletions made based on the ARE distribution. FL: full-length; Del: Deletion. **c** *Fxr1* knockdown promotes the luciferase activity-mediated by both 3′-UTRs of *p21* and *Fbxo4* mRNAs in 74B cells. **d** *Fxr1* overexpression suppresses the luciferase activity-mediated by 3′-UTR of *Fbxo4* mRNA in HEK293T cells. **e** ARE deletion rescues luciferase activity in HEK293T cells with *Fxr1* overexpression. **f** ARE deletion reduces luciferase activity in 74B cells with *Fxr1* knockdown. **g** The comparison of basal luciferase activity with full-length and deleting 3′-UTRs of *Fbxo4* mRNA in 74B cells. **h** Proposed model summarises the regulation of Fxr1 by Fbxo4 and feedback regulation of Fbxo4 by Fxr1. All the data represent mean ± s.d. **c**, **d** were analysed by Student's *t* test ($n = 3$). **e**–**g** were analysed by One-way ANOVA, followed by Fisher's LSD as post hoc test ($n = 3$). **\*\***$p < 0.01$

$\alpha$[26, 27]. Bioinformatic analyses revealed ARE elements in *Fbxo4* mRNA (Fig. 7a and Supplementary Data 11) (http://nibiru.tbi.univie.ac.at/AREsite2/welcome). *Fbxo4* mRNA and protein were collected from cells with either *Fxr1* knockdown or overexpression. *Fxr1* knockdown elevated Fbxo4 protein levels, while *Fbxo4* mRNA was either reduced or remained constant (Fig. 6a, b). In contrast to knockdown, *Fxr1* overexpression reduced Fbxo4 protein without reducing *Fbxo4* mRNA (Fig. 6c, d). We therefore assessed Fxr1-dependent control of Fbxo4 translation through polysome profiling in NIH3T3 cells with or without Fxr1 overexpression. A shift of *Fbxo4* mRNA towards monosome was detected in cells with *Fxr1* overexpression (Fig. 6e–g), consistent with Fxr1 antagonising *Fbxo4* translation. The presence of ARE elements in the *Fbxo4* 3′-UTR suggested that regulation could be direct. Consistent with direct regulation, Fxr1 binding to *Fbxo4* mRNA was detected by RIP in both 74B (Fig. 6h, i and Supplementary Fig. 11h) and NIH3T3 cells (Supplementary Fig. 11i, j).

To further examine the feedback regulatory loop, luciferase reporters with the *Fbxo4* 3′-UTR were constructed (Fig. 7a, b). *Fxr1* knockdown-induced luciferase activity-mediated by *Fbxo4* 3′-UTR (Fig. 7c); conversely, overexpression of Fxr1 suppressed expression (Fig. 7d). To define which AREs exert control of *Fbxo4* translation, deletions were constructed based on the distribution of AREs in *Fbxo4* 3′-UTR (Fig. 7b). Luciferase reporter assays revealed that all these three regions of AREs contribute to Fbxo4 regulation (Fig. 7e, f). In 74B cells with high levels of endogenous Fxr1, a rescue of basal luciferase activity was detected by deletion of ARE elements (Fig. 7g). Taken together, these data support a self-amplifying, regulatory feedback loop, wherein reduced Fbxo4 function results in Fxr1 overexpression due to reduced degradation (Fig. 7h). Fxr1 overexpression inturn reduces Fbxo4 levels, leading to a further reduction in E3 ligase activity and increased expression of Fxr1 itself.

## Discussion

Ubiquitin-dependent protein degradation provides a critical barrier that prevents overexpression and dysregulation of a majority of cancer drivers. As such, E3 ligases, which direct substrate specificity, often exhibit tumour suppressive functions and are inactivated during neoplastic transformation. Critical examples include members of the F-box family, such as Fbxw7 and Fbxo4[28, 29]. Fbxw7 directs proteolysis of key pro-neoplastic proteins, including c-Myc, Notch and cyclin E[30]. Mutations in the gene encoding *Fbxw7* occur in a number of malignancies and *Fbxw7*-deficient mice are tumour prone, demonstrating its tumour suppressive function[31, 32]. Fbxo4 likewise regulates degradation of cyclin D1[3, 4, 6], a key cancer driver and the DNA binding protein, Trf1/Pin2, a component of the telomere-capping complex[33]. *Fbxo4* is subject to both inactivating mutations and reduced expression in cancers[3, 14, 22]. The nature of Fbxo4 downregulation has to this point not been addressed. Although cyclin D1 has been biochemically and genetically demonstrated to be a key SCF$^{Fbxo4}$ substrate, it is unlikely to be the sole, biologically substrate. To identify additional substrates of SCF$^{Fbxo4}$, we utilised a proteomics approach to identify additional substrates. Using this approach, we noted co-purification of FMRP family proteins, FMRP, Fxr1 and Fxr2. Binding and direct ubiquitylation of Fxr1 validated it as an SCF$^{Fbxo4}$ substrate, while FMRP and Fxr2 could not be confirmed. Rather, the observed co-precipitation of FMRP likely reflects its ability to heterodimerise with Fxr1 (Supplementary Fig. 1e). Consistently, FMRP ubiquitylation is catalysed by the anaphase-promoting complex, where FMRP recognition is mediated by the Cdh1 subunit[34]. At physiological conditions, *Fxr1* is expressed in brain and muscle[35, 36]. Although highly homologous to *FMRP*, and thus expected to play a key role in neuronal homeostasis, Fxr1 function in fragile X syndrome remains ambiguous[37]. Critically, with regard to cancer, Fxr1 is overexpressed at a high frequency in head and neck cancers[12], suggesting it has unique substrates and functions.

Ubiquitylation-mediated protein degradation is precisely and tightly controlled in normal cells in a timely manner and in a specific location in order to maintain cellular homeostasis[38]. Timing and selectivity can be determined through regulation of ligase activity or by substrate modification; for example, phosphorylation of either the E3 ligase or the substrate itself[39]. Although SCF E3 ligases are generally considered to be constitutively active and their catalytic activity depends solely on modification of substrates that marks them for degradation, the activity of SCF$^{Fbxo4}$ is dependent on phosphorylation by GSK3$\beta$[3, 40–42]. This phosphorylation is necessary for Fbxo4 dimerisation and activation[3, 22]. Consistently, Fxr1 ubiquitylation requires phosphorylation-dependent activation of Fbxo4.

The binding of many SCF ligases to substrates requires substrate phosphorylation, including recognition of cyclin D1 by SCF$^{Fbxo4}$. Importantly, SCF$^{Fbxo4}$ can also bind to non-phosphorylated substrates, such as, Trf1. Fxr1 represents the second substrate in the latter category. Fxr1 can be phosphorylated by different kinases like Pak1 at Ser420, and Erk2 and GSK3$\beta$ at multiple sites[16, 43]. However, all of these sites are located within the C-terminus that is outside of the binding domain defined in our work. Our own phospho-Mass Spectrometry failed to define additional GSK3$\beta$ sites within or near the Fbxo4-binding domain, but did confirm previously reported sites. While a consensus degron that distinguishes phosphorylation-dependent (cyclin D1) from non-phosphorylation-dependent substrates (Trf1 and Fxr1) remains incompletely defined, the identification and characterisation of more substrates should facilitate the systematic identification of such motifs.

An additional level of proteolytic control concerns substrate versus ligase subcellular localisation; a classic example is cyclin D1. Cyclin D1 functions primarily in the nucleus, where as an activator of CDK4, it initiates phosphorylation-dependent inactivation of Rb[44]. However, at the G1/S boundary, it undergoes phosphorylation-dependent nuclear export; once in the cytoplasm, it is recognised by SCF$^{Fbxo4}$, which itself is a cytoplasmic complex. Additional examples of such regulation by differential subcellular localisation include p27$^{Kip1}$ and NEMO/IKK[39]. Fxr1 is distributed between the nucleus and cytoplasm. The nature of this regulation and whether it contributes to cell-cycle-specific degradation remains to be examined. It remains plausible that ubiquitylation and degradation of cytoplasmic Fxr1 is limited by GSK3$\beta$-dependent activation of SCF$^{Fbxo4}$ and nuclear Fxr1 is regulated independent of SCF$^{Fbxo4}$.

Fxr1 can interact with AGO2 to form microRNA–protein complexes that activate the transcription of downstream targets, such as, *TNF-$\alpha$* and *Myt1*[26, 45]. Fxr1 can also form a complex with PRKCI and ECT2, which controls cell proliferation/cell survival and links Fxr1 expression with human tumours; of note, elevated *Fxr1* mRNA levels correlate with poor prognosis[11]. With regard to its function in tumorigenesis, Fxr1 facilitates the bypass of senescence via suppressing *p21$^{Cip1}$* expression and stabilisation of *TERC* mRNA[12]. Fxr1 may have more roles in tumours and it is a reasonable target for investigation.

Although gene amplification contributes to Fxr1 overexpression in cancers[11, 12], we now demonstrate that post-translational regulation of Fxr1 is also a contributing factor. Our results reveal a critical role of Fbxo4 in maintaining homeostatic Fxr1 levels. Loss of Fbxo4 directly contributes to Fxr1 overexpression in both normal and cancer cells; likewise, re-

introduction of Fbxo4 into HNSCC cells triggers Fxr1-dependent senescence, demonstrating the importance of this regulatory loop. Rescue experiments support the importance of both p21[Cip1] and p27[Kip1] as downstream factors that control cell senescence and proliferation upon *Fxr1* knockdown or *Fbxo4* overexpression. Mechanistically, the regulation of p21[Cip1] by Fxr1 is clear, however, further studies are required to investigate how p27[Kip1] is regulated by Fxr1.

One striking result stemming from our consideration is the mechanism of Fbxo4 loss in HNSCC. Although *Fbxo4* is subjected to mutations in ESCC, the mutation frequency is much lower than that with protein loss or reduction in primary HNSCC (Supplementary Fig. 11). During the course of our molecular analysis, we noted that Fbxo4 protein expression fluctuated inversely with Fxr1 levels, suggesting Fbxo4 expression could be regulated by Fxr1, and a potential feedback amplification loop is established. Molecular analysis revealed that Fxr1 could directly bind to *Fbxo4* mRNA. Fxr1 regulates gene expression through either destabilisation of mRNA or inhibition of protein translation. Since we noted that *Fbxo4* mRNA was not influenced by Fxr1, we assessed *Fbxo4* translation by polysome analysis. These experiments reveal that Fxr1 inhibits *Fbxo4* mRNA enrichment in the actively translating polysomes. Our data support a model, wherein *Fxr1* amplification may be the initial hit. The increase in Fxr1 will suppress SCF[Fbxo4] function, triggering further Fxr1 elevation, resulting in an 'auto-amplifying loop' (Fig. 7h). The suppression of SCF[Fbxo4] function has additional consequences, for example, increased expression of other pro-neoplastic substrates. In this manner, the modest increase of *Fxr1* expression will have a profound impact on tumorigenesis by virtue of its capacity to suppress a documented tumour suppressor, *Fbxo4*, resulting in overexpression of pro-neoplastic SCF[Fbxo4] targets. It is also worth noting the potential for additional compensatory mechanisms that may facilitate Fxr1 downregulation by SCF[Fbxo4] E3 ligase, which would further enhance the pro-tumorigenic activity of Fxr1 in HNSCC. Our work demonstrates that Fxr1 is an SCF[Fbxo4] substrate, while the feedback regulation by Fxr1 reveals a broader than anticipated contribution of Fbxo4–Fxr1 axis to neoplastic progression.

## Methods

**Human tissues and TMAs.** Human tumour and adjacent normal tissues were collected from the Biorepository & Tissue Analysis at Hollings Cancer Centre with both written informed consent and MUSC Internal Review Board approval (Pro00009235 (CT)#101547). Frozen tumour and normal tissues were applied for Western blot. The TMAs with both HNSCC and normal tissues were obtained for Fbxo4 and Fxr1 IHC staining.

**Animal models and MEFs maintenance.** Mouse breeding, genotyping, handling and treatment were carried out in accordance with IACUC protocols and University Laboratory Animal Research guidelines at the Medical University of South Carolina. *Fbxo4* + / + , + /− and −/− transgenic mice in C57BL/6 background were developed by Vega Biolab (Philadelphia, PA). Six-week old male and female mice were utilised for breeding. At menstrual age day-14, mouse embryos were dissected out. The head and visceral organs and tissues were removed. Cells were maintained in MEFs medium on a 3T9 passaging protocol. MEFs medium contains Dulbecco modified Eagle's medium (DMEM) with 10% foetal bovine serum (FBS) (Gemini Bio-Products), 2 mM glutamine, 0.1 mM nonessential amino acids, 55 μM β-mercaptoethanol, and 10 μg of gentamicin/ml. The genotyping primers were as follows: 1loxP forward, 5′-GGCAGAGCTTGAGTTTGCAACATTTCAGGTG-3′, and 3loxP reverse, 5′-TCCTGATCTTTGGAAATTCTTCCTCTGAGT-3′.

**Cell culture.** HEK293T and NIH3T3 cells were cultured in DMEM containing 10% FBS and 1% penicillin-streptomycin. U2OS cells were maintained in McCoy's 5 A medium with 10% FBS and 1% penicillin-streptomycin. 74B cells were maintained in DMEM with 10% FBS and 1% penicillin-streptomycin; 74A cells were maintained in 74B medium plus 1% NEAA; SCC9 cells were cultured in DMEM:F12 (1:1) containing 400 ng/ml hydrocortisone, 10% FBS, and 1% penicillin-streptomycin. TE7, TE8, TE10 and TE15 ESCC cells were cultured in RPMI1640 plus 10% FBS and 1% penicillin-streptomycin. 451Lu, WM88, WM983B, WM3918,

1205Lu and WM793B melanoma cells were propagated in Tu2% medium: 80% MCDB153, 20% Leibovitz's L-15, 2% FBS, 4 mM Glutamine, and 1.68 mM CaCl₂. All cells were maintained in humidified incubator with 5% CO₂.

**Western blot and immunoprecipitation.** After treatment, cells were collected and lysed in RIPA buffer (50 mM Tris pH7.5, 150 mM NaCl, 1.0% NP-40, 0.1% SDS and 0.5% Deoxycholic acid) supplemented with protease and phosphatase inhibitors. Lysates were resolved in SDS-PAGE gels for western blotting. HEK293T cells were transiently transfected with interested plasmids; 24 h post-transfection, cells were lysed in Tween 20 buffer. The same amounts of cell lysate were applied in immunoprecipitation using Flag affinity gel, anti-c-Myc agarose affinity gel (A7470, Sigma-Aldrich) or anti-His affinity resin (L00439, GenScript). After washing, beads were boiled at 95 °C for 5 min and immunoblotting was performed. Representative full blot images are shown in Supplementary Fig. 12.

**Homology modeling of Fxr1.** Using the software MOE from Chemical Computing Group Inc.[46], a homology model of the amino-terminal of Fxr1 (aa1–aa207) was created based on X-ray-derived coordinates of the FMRP amino-terminal domain, PDB:4QVZ.B, which contains a tandem Tudor and KH motif[47]. Fxr1 was first aligned to FMRP based on the BLOSUM64 matrix. The overall similarity between proteins was 59%, but within the amino-terminal domain used for the homology model, the identity was 81% (Supplementary Fig. 2b). For homology modelling, the structural file was corrected for missing atoms based on the amino acid sequence, the missing loop between A[98]–T[102] was built using a rotamer library, and finally the protein was protonated at $T = 310$ K, pH 7.3, salt at 200 mM, and using GB/VI electrostatics. Homology modeling produced ten intermediate, which were scored based on the electrostatic solvation energy, the structures were energy minimised using the AMBER12:ETH force field, and the final model was determined based on the best electrostatic solvation energy.

**Bimolecular docking of Fbxo4 with Fxr1.** After creating the structural file for Fxr1, the Fbxo4 X-ray PDB:3L82.B bimolecular docking of the carboxyl terminus was performed[7]. The 3L82.B molecule was truncated at the amino terminus to begin at reside P[177], and the Fxr1 homology model was truncated at the carboxy terminus to end at residue L[198] due to the extended amino or carboxy terminus creating an overhang that behaves like a decoy-promoting artifacts in simulations. This can be seen when comparing Fbxo4 in Fig. 1c vs. Fig. 1e. Using the docking server ClusPro[13, 48], the Fxr1 structure was used as the ligand and the Fbxo4 structure was used as the receptor. Briefly, ClusPro uses a rigid body docking algorithm to test ten billion possible spatial combinations of the protein pair, then through iterative calculations of shape complementary and electrostatics, the top 2000 protein pair poses are selected and grouped into clusters based on root mean square deviation (RMSD ≤ 10 Å) of the overall poses. The poses that are considered best are determined by the number of poses in each cluster and the pose scores (based on shape complementary and electrostatics). In the case of Fbxo4 and Fxr1, the binding residues were not restricted nor were any residues selected to block docking and default settings were used to allow maximum freedom of docking poses. The top 10 best poses using the balanced coefficient weighting from the output were analysed (Supplementary Fig. 3) and the best consensus pose (C1_243) is presented (Table 1). Ten.pdb files are provided as Supplementary Data 1–10 named after the ClusPro rank and number of poses in the model cluster, in which Chain A is Fbxo4 and Chain B is Fxr1.

**Molecular interaction analysis.** The heterodimer structures from PDB:3L82 and the best heterodimer output from ClusPro for Fbxo4:Fxr1 were interrogated for intermolecular contacts. Protein contact thresholds were 4.5 for hydrophobic interactions, 4.2 for ionic bonds, 2.5 for disulphide bonds with a sequence separation of 4 and a network separation of 0. Molecular images were prepared using MOE.

**Sequence analysis.** The amino acid sequences were analysed by pairwise aligning using the BLOSUM64 matrix, and were also visualised using BioEdit 7.2.5. For the pairwise alignment of Fxr1:Trf1, the identities were 17% and similarities were 33% for the full proteins, but increased within the region of interest.

**Senescence staining.** HNSCC cells were transiently infected with Con vector, WT *Fbxo4*, *Fxr1* shRNA, and WT *Fbxo4* plus *Fxr1* in 35 mm dishes. 72 h later, SA-β-gal activity is determined using X-gal (5-bromo-4-chloro-3-indolyl β-D-galactoside) staining at pH 6.0 according to the manufacturer's instructions (9860, Cell Signalling Technology and CS0030, Sigma-Aldrich).

**Immunohistochemistry.** Paraffin-embedded sections were microwaved, blocked and incubated with primary antibodies (dilution: human TMA - Fbxo4 (1:50) and Fxr1 (1:50) and mouse tissue - Fxr1 (1:150)), and signal was amplified using VECTASTAIN Elite ABC HRP Kit and detected by Vector® DAB Substrate. Following IHC staining, all sections were counterstained with hematoxylin, dehydrated, and mounted. Normal rabbit IgG was used as negative control[49]. After staining, the slides were reviewed blinded to original diagnoses. Staining index (SI) was assessed to quantify the expression of Fbxo4 and Fxr1. Ten high-power fields

were chosen randomly and evaluated. The average percentage of positively stained cells were scored by the positive range score: 0 = 0–10%; 1 = 11–30%; 2 = 31–70%; 3 = 71–100%. The positive intensity score reflected the colour: 0 = no staining; 1 = light yellow; 2 = yellow; 3 = brown. The SI equals the product of the positive range score and the positive intensity score.

**Polysome profiling**. NIH3T3 cells were infected with retrovirus control or that encoding *Fxr1*. 24 h post infection, cells were lysed in $TMK_{100}$ lysis buffer and the supernatant was layered onto a 10–50% sucrose gradient and centrifuged at 151,000×$g$ at 4 °C for 3 h. Polysome fractions were collected using a fraction collector with continuous monitoring of absorbance at 254 nM. RNAs were extracted with Trizol (Invitrogen) and reverse-transcribed to cDNAs using SuperScript III Reverse Transcriptase. PCR was performed using primers listed below: *Fbxo4*: 5′-TCAACAGCAACTCCCACTCTTCCA-3′ and 5′-ACCCTGTTGCTGTAGCCGT ATTCA-3′; *GAPDH*: 5′-GTTGATGTGCAGTTGTATATCTTGTC-3′ and 5′-GCG TATATGGACAGCACATTTTATAA-3′. Two percent agarose gel was utilised to resolve the PCR products. Band quantification was performed using Quantity One (Bio-Rad Laboratories, Inc.).

**RIP analysis**. RIP was performed using the commercial available kit: Magna RIP™ RNA-Binding Protein Immunoprecipitation Kit (17–700, EMD Millipore). An equivalent amount of 74B or NIH3T3 cell lysate were used for immunoprecipitation with Fxr1 antibody and normal mouse IgG control (03–176, EMD Millipore) for 3 h. After immunoprecipitation, RNA was extracted with Phenol:chloroform: isoamyl alcohol (125:24:1, pH = 4.3), and chloroform. Finally, RNA was reverse-transcribed using SuperScript III Reverse Transcriptase, and quantified by semi-quantitative PCR.

**Statistical analysis**. Plots were made either by GraphPad Prism6 or Microsoft Excel 2011. Statistical analyses were performed using IBM SPSS Statistics 24. The values are shown as mean ± s.d. For statistical analysis Student's $t$ test, Mann–Whitney $U$ test, Kruskal–Wallis Test, One-way ANOVA, Two-way ANOVA and $\chi^2$ test were used to compare the data. The results with $p$ values <0.05 are considered significance.

**Data availability**. The authors declare that all the data supporting the findings of this study are available within the article and its Supplementary Information files and from the corresponding author on reasonable request.

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

## Acknowledgements

This work was supported by grants from the NIH: P01 CA098101 (J.A.D.), R01CA093237 (J.A.D.); R01DE022776 (V.P.). The authors wish to thank Dr Hui-Kuan Lin for providing His-tagged Ubiquitin WT, K48R and K63R plasmids. We thank the members of Dr Diehl lab for critical insight.

## Author contributions

J.A.D., S.Q., V.P.: Designed and interpreted the experiments. S.Q. and M.M.: Performed the experiments. J.A.D., S.Q., V.P.: Wrote and edited the manuscript. K.M.: Assisted with IHC. Y.K.P.: Performed the molecular modeling. B.V.H. and P.H.H. Assisted with polysome profiling.

## Additional information

**Competing interests:** The authors declare no competing financial interests.

