## [Peer Review File · Nature Communications]

Reviewers' comments:

Reviewer #1 (Remarks to the Author):

In this manuscript, the Diehl laboratory describes the identification of the RNA-binding protein Fxr1 as a ubiquitination substrate of the E3 ligase complex Fbxo4-Skp1-Cul1-F-box. The authors show in detail the inverse relationship between Fbxo4 and Fxr1 levels in cultured cells as well as in head and neck squamous cell carcinoma. The authors then identify an interesting feedback regulatory loop whereby Fxr1 binds Fbxo4 mRNA and represses Fbxo4 translation.

The area investigated –ubiquitin-mediated control of tumor suppressor proteins— is important and timely. In addition, the discovery that Fxr1 is regulated via ubiquitin-mediated proteolysis by the Fbxo4-SCF complex is novel and potentially exploitable for therapy. One arm of the model, the regulation of Fxr1 expression by Fbxo4 is characterized in thorough and complete experiments. Unfortunately, the other arm of the model, the regulation of Fbxo4 expression by Fxr1, is rather preliminary. The experiments that connect the Fbxo4-Fxr1 axis to senescence and proliferation also need to be strengthened.

Main comments

In Figure 5, the authors find connections between Fxr1, the suppression of senescence, and the promotion of proliferation. The authors find that Fxr1 knockdown leads to increased levels of p21 and p27 and strong activity of the senescence marker beta-galactosidase, while Fxr1 overexpression suppressed p21 and p27 production. The authors need to complete these results in two areas: (1) They need to show that Fxr1 binds to p21 mRNA and p27 mRNA in HNSCC. In fact, there does not appear to be prior data showing that Fxr1 binds p27 mRNA or influences its post-transcriptional metabolism. Thus, the authors need to establish whether Fxr1 lowers the stability of p21 mRNA and/or p27 mRNA. They also need to test if additionally (or instead), Fxr1 lowers the translation of p27 mRNA or both mRNAs.

(2) The authors must employ rescue experiments to test if p21 and p27 are direct downstream effectors of the actions of Fxr1 on senescence and proliferation. Similarly, they must test if p21 and p27 mediate the actions of Fbxo4 on senescence and proliferation using rescue experiments.

In Figure 7, the authors focus on the regulation of Fbxo4 expression by Fxr1. Given the novelty of this interaction and the fact that it is central to the authors' model, key structural and functional details of this complex need to be provided:

(3) The site of interaction of Fxr1 on Fbxo4 mRNA needs to be mapped and demonstrated experimentally.

(4) After they have found the site(s) on the Fbxo4 mRNA that interacts with Fxr1, heterologous mRNA reporter constructs (e.g., luciferase) must be employed to verify that the impact of Fxr1 on Fbxo4 mRNA is direct.

(5) The consequences of silencing Fxr1 on Fbxo4 translation (at a minimum by monitoring Fbxo4 mRNA distribution on polysome gradients) must be tested.

Reviewer #2 (Remarks to the Author):

The manuscript discusses SCF(Fbxo4) interaction with FXR1, and elucidates the role of this interaction in normal and cancer cells (when Fxr1 is overexpressed).

My comments are mostly related to structure modeling aspects of the interacting complex. The

authors build a model of Fbxo4 FXR1 interaction. Current PDB contains crystal structure of Fbxo4 (also the same protein complex with Trf1), and a close homologue of FXR1 (FMRP ~70%). Since Trf1 is not similar to FRMP (and hence FXR1) the authors used protein docking to generate potential models of interaction. After that, they attempted to validate the interaction models by using alanine scanning. The overall proposed protocol sounds reasonable, however it needs some further clarification. Below are my suggestions

Major:

The authors claim that most of the docking models went to the interaction site of Trf1, which together with mutation I375M sounds convincing for the Fbxo interaction site, however Fxr1 site is more uncertain. The authors should more clearly describe what they did. Because in current writeup instead of discussing potential model interface for choosing interface residues, the authors started to use very remote sequence similarity to Trf1. (which is not very certain since the proteins are quite different) It would be more logical to analyze docking models and see whether some of those match the conserved regions in the alignment. Even though I'm not fully convinced that Trf1 and Fxr1 should interact similarly. Ideally performed mutations should validate one of the docking models vs the others. Also docking model (perhaps analyzed models) should be provided in the supplement (in pdb format).

Minor:

The authors should mention that they use docking in the main part of the manuscript, since in the current writeup it is not obvious what exactly was done, when reading the main paper.

The way it is written one can think that the authors use template based complex building which is definitely not the case as seen on the Figure 1, which causes confusion

Also docking protocol should be moved to the methods of the main paper, since interpretation of those are non-trivial.

Reviewer #3 (Remarks to the Author):

In the current paper Qie and colleagues report that the SCF-FBXO4 ubiquitin ligase inhibits oncogenesis in head and neck squamous cell carcinoma by targeting FXR1 for degradation. The authors demonstrate that the F-box protein FBXO4 interacts with FXR1 and controls FXR1 turnover by promoting its ubiquitylation. They also show that FBXO4 controls proliferation and senescence of head and neck squamous cell carcinoma cells in an FXR1-dependent fashion. Moreover, they report that the decreased levels of FBXO4 observed in primary head and neck squamous cell carcinoma correlate with FXR1 overexpression.

In my opinion this study is potentially interesting, however, several conclusions are not supported by the data presented. Below I provide ways this study can be strengthened.

1. Figure 1E. It is shown that the FBXO4 mutants in which Glu379 and Glu380 are mutated to alanine do not interact with FXR1, however, the results shown in this figure are not convincing. Instead of overexpressing both FBXO4 (wild type or mutants) and FXR1, the authors should pull down overexpressed FBXO4 wild type, FBXO4-E379A or FBXO4-E380A and test whether they coimmunoprecipitate with endogenous FXR1.

2. Figure 1H. It is stated that the V178A mutation of FXR1 disrupts its binding to FBXO4. However, the

amount of immunoprecipitated bait (MYC-epitope tagged FXR1) mirrors the one of FBXO4 that is pulled down (compare lanes 1 and 2). The authors should try to overexpress and pull down the different FXR1 mutants and test their ability to immunoprecipitate endogenous FBXO4. More generally, the statement "AA172-200 mediates FBXO4 recognition" is not supported by the authors' data.

3. Figure 2B. Why does MLN-4924 treatment lead to accumulation of FXR1 even in the absence of FBXO4 (lanes 3 and 4)? The FXR1 increase caused by MLN-4924 is similar in FBXO4+/+ and FBXO4-/- MEFs (compare lanes 1 and 2 with lanes 3 and 4). How is this result explained?

4. Figure 5G-I. The quantification of beta-Gal-positive senescent cells is missing.

5. Is the FXR1 binding to FBXO4 specific for this F-box protein? Have the authors verified that FXR1 is not pulled down by other FBPs?

Minor:

Figures 2 and 3 should be combined and some panels moved to supplemental information.

Reviewer #4 (Remarks to the Author):

This manuscript described that a novel substrate of Fbxo4, Fxr1 and their role in head and neck squamous cell carcinoma. In general, the study is well-designed and performed. The results are convincing, and the discussion are appropriate. Below are comments to individual figures:

Figures 1-3:

First, the experiment is done properly and solidly. The demonstration of interaction between Fxr1 and Fbxo4. Co-immunoprecipitation by using Flag-Fbxo4 or Flag-Fbxo4 Δ F constructs, plus co-expressing myc-Fxr1, cyclin D1 (as positive control) is a sufficient experimental strategy. Further, data presented in the later part of this section is sufficient and thoroughly identify and prove the critical molecular basis of interaction between Fbxo4 and Fxr1. Relevant experiments are done properly and sufficient by assessing multiple Fbxo4 mutants, including almost all critical mutants within Fbxo4 that have been previously reported. Additionally, multiple experiments and sequencing analyses were also solidly performed to diminish the possibility that other Fragile X mental retardation syndrome proteins family members may also function as putative substrates of Fbxo4, lay the rationale and foundation for the sole investigation into Fxr1.

Further, the study also sufficiently proves Fxr1 is a putative substrate of Fbxo4 in HEK293T cells. First, it is superior to show molecular/genetic or pharmacological intervention of Fbxo4 didn't alter Fxr1 gene expression but protein level. Second, multiple comprehensive biochemical assessments were conducted to show that SCFFbxo4 directly ubiquitylates and degrades Fxr1 that maybe in a manner that depends on GSK3 β -phosphorylated Fbxo4. Considering the previously established role of GSK3 β to phosphorylate Fxr1 for down-regulation, the study also used multiple firmly-designed experiments to show co-expressing GSK3 β increase Fxr1 ubiquitylation in a SCFFbxo4 dependent manner, further detailing the molecular basis of such interaction. Considering the possibility that some unidentified components within SCFFbxo4 may also mediate Fxr1 recognition, the study also carefully used recombinant wild-type/mutant/ phospho-mimetic SCFFbxo4 protein to demonstrate a role of Fbxo4 phosphorylation in Fxr1 ubiquitylation.

In general, data from these sections are clear, sufficient, accurate, and compelling. However, considering GSK3 β targets multiple potent downstream factors that involve mechanisms such as supporting translation which could be a mediator of Fxr1 protein level, it may not be sufficient to

demonstrate "GSK3 β -dependent Fbxo4 phosphorylation" by simply overexpressing a sole wild type GSK3 β . Further experiments by using GSK3 β critical mutants may be needed to firmly prove this mechanism.

In addition, it would be more compelling (may not be that critical) to conduct a few simple experiments to demonstrate whether all those genetic-interfering cells also showed attenuated translation efficiency. In this study, assessing of Frx1 protein level is simply the final readout; however, it is the reduced protein level that may be affected not only by degradation, but may also be due to reduced translation efficiency.

Figure 4:

Next, the study steps into investigating if such mechanism also exists in head neck cancer cell models. Data presented in this section is generally compelling but not sufficient. First, by simply looking at gene expression of Frx1 in cancer cohorts may not be sufficient as this mechanism is completely established on protein level. It may be more compelling to move some data from Figure 6 into Figure 4 to further demonstrate Frx1 protein is elevated, in general, in head and neck cancer tissue samples. Second, the paper didn't specifically state which HNSCC cell line is used for knockdown (Fbxo4) experiment and failed to present relevant data of knocking-down efficiency. If knocking down effect is not assessed by using w.t. Fbxo4-expressing cells, it would be more compelling to perform a rescue experiment by applying recombinant w.t. Fbxo4 protein into the knocking down experiment. Third, although it is not critical, it is worthy to note that changes of Frx1 protein level in multiple data sets shown in Figure 4 are not compelling (considering the change of W.B band intensity), may bring a question that a compensation mechanism may raise due to genetic alteration in head and neck cancer cells. This needs to be discussed and addressed. Fourth, as such mechanism involves "GSK3 β -dependent Fbxo4 phosphorylation", it would be more compelling to show the "GSK3 β -dependent Fbxo4 phosphorylation" also exists in head and neck cancer cells. As GSK3 β is such an important cancer-associated factor that crosses with many critical signaling such as PI3K that regulates various cell functions including cell proliferation and senescence, it would be interesting to see whether interfering GSK3 β also affect Frx1 protein level and relevant phenotypes. Many GSK3 β inhibitors are available. It only needs a simple experiment by treating such inhibitor in various already-established vector-expressing cells to assess Frx1 protein level.

Figures 5-7:

The study further investigated whether Fbxo4 regulates head and neck cancer cell proliferation and senescence. From the data (Fig 5. A/B/C), it seems that the effect of Fbxo4 knockdown on cellular proliferation only observed after day-3, bringing a question whether cells are confluent at day 3 and the difference only observed when cells reached to confluency and the effect is more dependent on a cell-to-cell contact manner, rather than the putative cell growth. The paper didn't state/discuss this. It needs to be more clarified, especially in the context to assess senescence which is highly dependent on cell confluency in an experimental perspective. Once again, the knockdown efficiency in relevant cells are not presented in the data set, which bring some smoke to assess the putative effect, especially multiple experiments are done by concurrent knocking down of both Fbxo4 and Frx1 and in a scenario that knocking down of Fbxo4 itself enhances Frx1 protein expression. In addition, for the β -Gal staining assay (Fig.5 G/H/I), it is also recommended to show a quantification data by assessing multiple bright-field records, not only showing a few representative images. In addition, it would be wise to also clarify the experimental conditions for these experiments, whether the experiments were done by using selection agent such as puromycin to force the knockdown efficiency as sometimes, selection agent would promote cellular senescence at different levels. Finally, the data (Fig.5 G/H/I) suggest that "Fbxo4 expression increased SA- β -Gal staining to a similar degree as Frx1 knockdown", which brings some concerns that the reduced Frx1 protein level is due to Fbxo4/GSK3 β -mediated degradation or simply due to Fbxo4-induced senescence, which translation could be halted. It is recommended to design and perform additional experiments such as polysome profiling to rule out the

possibility that the reduced Fxr1 protein expression is due to the senescence. Otherwise the study is disconnected between the molecular studies and the cancer cells – related studies, regardless correlation between Fbxo4 loss and Fxr1 overexpression in HNSCC tumors and relevant mouse models presented in Figure 6.

Reviewers' comments:

Reviewer #1 (Remarks to the Author):

In this manuscript, the Diehl laboratory describes the identification of the RNA-binding protein Fxr1 as an ubiquitination substrate of the E3 ligase complex Fbxo4-Skp1-Cul1-F-box. The authors show in detail the inverse relationship between Fbxo4 and Fxr1 levels in cultured cells as well as in head and neck squamous cell carcinoma. The authors then identify an interesting feedback regulatory loop whereby Fxr1 binds Fbxo4 mRNA and represses Fbxo4 translation.

The area investigated –ubiquitin-mediated control of tumor suppressor proteins— is important and timely. In addition, the discovery that Fxr1 is regulated via ubiquitin-mediated proteolysis by the Fbxo4-SCF complex is novel and potentially exploitable for therapy. One arm of the model, the regulation of Fxr1 expression by Fbxo4 is characterized in thorough and complete experiments. Unfortunately, the other arm of the model, the regulation of Fbxo4 expression by Fxr1, is rather preliminary. The experiments that connect the Fbxo4-Fxr1 axis to senescence and proliferation also need to be strengthened.

Main comments

In Figure 5, the authors find connections between Fxr1, the suppression of senescence, and the promotion of proliferation. The authors find that Fxr1 knockdown leads to increased levels of p21 and p27 and strong activity of the senescence marker beta-galactosidase, while Fxr1 overexpression suppressed p21 and p27 production. The authors need to complete these results in two areas:

(1) They need to show that Fxr1 binds to p21 mRNA and p27 mRNA in HNSCC. In fact, there does not appear to be prior data showing that Fxr1 binds p27 mRNA or influences its post-transcriptional metabolism. Thus, the authors need to establish whether Fxr1 lowers the stability of p21 mRNA and/or p27 mRNA. They also need to test if additionally (or instead), Fxr1 lowers the translation of p27 mRNA or both mRNAs.

Reply: With regard to p21, direct binding of Fxr1 to p21 mRNA is confirmed in our paper (Supplementary Fig. 11h,j) and previously published in PLoS Genet. 2016 Sep 8;12(9):e1006306, in which it showed Fxr1 can bind to and promote the degradation of p21 mRNA.

With respect to p27, (Figure 2 A & G) results in our PLoS Genet paper demonstrate that Fxr1 knockdown increases p27 mRNA levels, however, no binding between Fxr1 and p27 mRNA was detected by RIP analysis. We have confirmed that Fxr1 overexpression reduces p27 protein levels in NIH3T3 cells (this manuscript, Fig. 6c). Collectively, these data reveal that p27 is indirectly regulated by Fxr1, while p21 is a direct target.

(2) The authors must employ rescue experiments to test if p21 and p27 are direct downstream effectors of the actions of Fxr1 on senescence and proliferation. Similarly,

they must test if p21 and p27 mediate the actions of Fbxo4 on senescence and proliferation using rescue experiments.

Reply: To directly answer this question, we utilized shRNAs to knockdown either p21 or p27 or both in 74B cells and assessed effects upon Fxr1 knockdown or Fbxo4 overexpression.

Western blots show the knockdown of p21 and p27 (Fig. 4h,j). Upon double knockdown of p21 and p27, cell senescence and proliferation can be rescued upon Fxr1 knockdown or Fbxo4 overexpression (Fig. 4e,f,g,i). As for single knockdown, p21 KD provides a stronger effect than p27 KD, consistent with the Western blot data, which shows a compensatory p21 upregulation upon p27 KD. Taken together, our data illustrate that both p21 and p27 are downstream factors of Fxr1 that control cellular senescence and proliferation; however, p21 appears to play the major role.

In Figure 7, the authors focus on the regulation of Fbxo4 expression by Fxr1. Given the novelty of this interaction and the fact that it is central to the authors' model, key structural and functional details of this complex need to be provided:

(3) The site of interaction of Fxr1 on Fbxo4 mRNA needs to be mapped and demonstrated experimentally.

Reply: We have added (Fig. 7a) and Supplementary Data 11 to demonstrate the existence of AREs in 3'-UTR of Fbxo4 mRNA. At the same time, we made luciferase reporters with WT or various deletions of the Fbxo4 3'-UTR to demonstrate that Fxr1 can regulate Fbxo4 translation directly through ARE motifs. Please also refer to answers to question (4) & (5).

(4) After they have found the site(s) on the Fbxo4 mRNA that interacts with Fxr1, heterologous mRNA reporter constructs (e.g., luciferase) must be employed to verify that the impact of Fxr1 on Fbxo4 mRNA is direct.

Reply: By doing the luciferase reporter assay, we found Fxr1 knockdown induces the luciferase activity-mediated by Fbxo4 3'-UTR (Fig. 7b); conversely Fxr1 overexpression suppresses its activity in an ARE-dependent fashion (Fig. 7c). To narrow down the AREs that control Fbxo4 translation, deletion mutations were constructed based on the distribution of AREs in Fbxo4 3'-UTR (Fig. 7a lower panel). Luciferase reporter assays demonstrated that all these three regions of AREs are important for Fbxo4 regulation (Fig. 7d,e). In 74B cells with high basal Fxr1 expression, the luciferase activity is rescued by all deletions (Fig. 7f), confirming Fxr1 directly regulates Fbxo4 translation.

(5) The consequences of silencing Fxr1 on Fbxo4 translation must be tested.

Reply: To address this question, we took advantage of the established luciferase reporter system. Luciferase assay was performed using Control cells and 74B cells with Fxr1 KD (Fig. 7b,e). Our results strongly suggest that Fxr1 directly regulates Fbxo4 translation through ARE elements within the 3'-UTR.

Reviewer #2 (Remarks to the Author):

The manuscript discusses SCF(Fbxo4) interaction with FXR1, and elucidates the role of this interaction in normal and cancer cells (when Fxr1 is overexpressed).

My comments are mostly related to structure modeling aspects of the interacting complex. The authors build a model of Fbxo4-FXR1 interaction. Current PDB contains crystal structure of Fbxo4 (also the same protein complex with Trf1), and a close homologue of FXR1 (FMRP ~70%). Since Trf1 is not similar to FMRP (and hence FXR1), the authors used protein docking to generate potential models of interaction. After that, they attempted to validate the interaction models by using alanine scanning. The overall proposed protocol sounds reasonable; however, it needs some further clarification. Below are my suggestions:

Major:

The authors claim that most of the docking models went to the interaction site of Trf1, which together with mutation I377M sounds convincing for the Fbxo4 interaction site; however, Fxr1 site is more uncertain.

Reply: We agree the Fxr1 site is more uncertain. This is also supported by our bimolecular docking predictions where the Fbxo4 interactions were highly conserved, but more diversity was found in Fxr1 interactions. To address this more clearly, we first improved the rationale and simplified the alignment in the manuscript to highlight the limited area of high conservation (Fig. 1e). Second, we improved and added details with regard to analyze the top models, which indicated a potential “hot spot” in Fxr1 that is supported by the protein alignments (Supplementary Fig. 3b).

The authors should more clearly describe what they did. Because in current writeup, instead of discussing potential model interface for choosing interface residues, the authors started to use very remote sequence similarity to Trf1. (which is not very certain since the proteins are quite different)

Reply: We agree the argument as present needs more detail as to the rationale. Our central hypothesis in terms of Fbxo4 interaction with Fxr1 is focused on a small conserved domain among mostly unrelated proteins. This domain is only part of the full interaction of the proteins.

It would be more logical to analyze docking models and see whether some of those match the conserved regions in the alignment. Even though I'm not fully convinced that Trf1 and Fxr1 should interact similarly.

Reply: This is an excellent suggestion and we fully agree with the reviewer. In fact, the suggestion was part of our protocol; however, it was not explained clearly in the first submission due to concerns with space constraints. We are not postulating that Trf1 and Fxr1 interact in exactly the same way, but share part of a similar binding motif that would work in concert with unique interactions for each pair. The

mutations were designed to test if this “hot spot” existed, but full elucidation of all required interactions was beyond the scope of this work.

After preparing the docking proteins and subjecting them to exhaustive bimolecular docking using ClusPro, the output was analyzed for consensus among top models and for which models indicated a potential interaction with the Fxr1 “hotspot” (aa173-aa192). We saw very strong consensus among 3/10 models and good similarity in 6/10 models (Supplementary Fig. 3). Of these models, the best model (C1_243) has the closest proximity of Fxr1 aa173-aa192 to Fbxo4 (Table 1). Another problem with bi-molecular docking is a propensity to find poses that use the termini of proteins and create artificial charge pairs that score well but are likely artifacts. These termini are not found in nature but arise from truncation of recombinant proteins in facilitating crystallization or a lack of resolution in the termini due to protein flexibility during crystallography. In our case models ranked 2 (C2_90), 4 (C4_60), 6 (C6_48), and 9 (C9_34) indicated the pose was driven by interactions with Fxr1’s termini and they are treated as suspect. With all these data in mind, the top model was chosen for representation in the manuscript in order to visualize the interaction for the reader. These details have been added to the manuscript for clarity (Supplementary Fig. 3 and Table 1).

Ideally performed mutations should validate one of the docking models vs the others.

Reply: This detail was not highlighted in the original submission. However, the S191A mutant had little effect on the interaction and supports the model since this residue is not predicted to form a bond with Fbxo4. The revision now highlights this detail. Unfortunately, the Fxr1 mutations that were performed were confined to single region of the protein, making differentiation between models, which all use different orientations, impossible. However, we think in the revision we make clear why the top model was a good pick with the addition of the expanded ClusPro results.

Also docking model (perhaps analyzed models) should be provided in the supplement (in pdb format).

Reply: We have submitted the models in pdb format as requested (Supplementary Data 1-10).

Minor:

The authors should mention that they use docking in the main part of the manuscript, since in the current writeup it is not obvious what exactly was done, when reading the main paper.

The way it is written one can think that the authors use template based complex building which is definitely not the case as seen on the Figure 1, which causes confusion. Also docking protocol should be moved to the methods of the main paper, since interpretation of those is non-trivial.

Reply: We have improved the text to do a better job capturing our rationale, how the in silico experiments were performed and, and integration of the in silico data with the cell based data.

We have moved the methods to the main text instead of as Supplementary Information.

Reviewer #3 (Remarks to the Author):

In the current paper Qie and colleagues report that the SCF-FBXO4 ubiquitin ligase inhibits oncogenesis in head and neck squamous cell carcinoma by targeting FXR1 for degradation. The authors demonstrate that the F-box protein FBXO4 interacts with FXR1 and controls FXR1 turnover by promoting its ubiquitylation. They also show that FBXO4 controls proliferation and senescence of head and neck squamous cell carcinoma cells in an FXR1-dependent fashion. Moreover, they report that the decreased levels of FBXO4 observed in primary head and neck squamous cell carcinoma correlate with FXR1 overexpression.

In my opinion this study is potentially interesting, however, several conclusions are not supported by the data presented. Below I provide ways this study can be strengthened.

1. Figure 1E. It is shown that the FBXO4 mutants which Glu379 and Glu380 are mutated to alanine do not interact with FXR1; however, the results shown in this figure are not convincing. Instead of overexpressing both FBXO4 (wild type or mutants) and FXR1, the authors should pull down overexpressed FBXO4 wild type, FBXO4-E379A or FBXO4-E380A and test whether they coimmunoprecipitate with endogenous FXR1.

Reply: We agree with this question; to directly answer it, we performed immunoprecipitation by only overexpressing Fbxo4 WT, E379A, E380A and I377M mutants in HEK293T cells and determined their interacting efficiency with endogenous Fxr1. Our data (Supplementary Fig. 4c) indicated that WT Fbxo4 shows a stronger capacity to interact with endogenous Fxr1 than the other mutants, supporting our initial conclusions and the molecular model.

2. Figure 1H. It is stated that the V178A mutation of FXR1 disrupts its binding to FBXO4. However, the amount of immunoprecipitated bait (MYC-epitope tagged FXR1) mirrors the one of FBXO4 that is pulled down (compare lanes 1 and 2). The authors should try to overexpress and pull down the different FXR1 mutants and test their ability to immunoprecipitate endogenous FBXO4.

More generally, the statement "AA172-200 mediates FBXO4 recognition" is not supported by the authors' data.

Reply: This is a very good question. To address it, we performed immunoprecipitation by only overexpressing Fxr1 WT and relative mutants in HEK293T cells (Supplementary Fig. 4e). As indicated by Western blot, V178A shows a lower binding capacity compared to WT and other mutants, highlighting the critical role of this site in mediating their interaction. We have changed the statement "AA172-200 mediates Fbxo4 recognition".

3. Figure 2B. Why does MLN-4924 treatment lead to accumulation of FXR1 even in the absence of FBXO4 (lanes 3 and 4)? The FXR1 increase caused by MLN-4924 is similar

in FBXO4^{+/+} and FBXO4^{-/-} MEFs (compare lanes 1 and 2 with lanes 3 and 4). How is this result explained?

Reply: Good question, we noted this result and thought the possibility of presence of additional E3 ligases that can also ubiquitylate and degrade Fxr1. We are currently pursuing this notion.

4. Figure 5G-I. The quantification of beta-Gal-positive senescent cells is missing.

Reply: We have already added the quantification of the percentage of senescent cells based on three biological repeats.

5. Is the FXR1 binding to FBXO4 specific for this F-box protein? Have the authors verified that FXR1 is not pulled down by other FBPs?

Reply: This is a real good question, we did not identify additional F-box proteins associated in our IP and MS analysis based analysis. However, this is a negative result and as such as mentioned for question No. 3, we are working to address/identify other E3 ligases that may also ubiquitylate and degrade Fxr1.

Minor:

Figures 2 and 3 should be combined and some panels moved to supplemental information.

Reply: This is a good suggestion. The combination can help improving the logic flow of this paper. To answer this, we have combined (Fig. 2 and Fig. 3) together and moved the additional data to (Supplementary Fig. 5 and Fig. 6).

Reviewer #4 (Remarks to the Author):

This manuscript described that a novel substrate of Fbox4, Fxr1 and their role in head and neck squamous cell carcinoma. In general, the study is well-designed and performed. The results are convincing, and the discussion is appropriate. Below are comments to individual figures:

Figures 1-3:

First, the experiment is done properly and solidly. The demonstration of interaction between Fxr1 and Fbxo4, co-immunoprecipitation by using Flag-Fbxo4 or Flag-Fbxo4 Δ F constructs, plus co-expressing myc-Fxr1, cyclin D1 (as positive control) is a sufficient experimental strategy. Further, data presented in the later part of this section is sufficient and thoroughly identify and prove the critical molecular basis of interaction between Fbxo4 and Fxr1. Relevant experiments are done properly and sufficient by assessing multiple Fbxo4 mutants, including almost all critical mutants within Fbxo4 that have been previously reported. Additionally, multiple experiments and sequencing analyses were also solidly performed to diminish the possibility that other Fragile X mental retardation syndrome proteins family members may also function as putative substrates of Fbxo4, lay the rationale and foundation for the sole investigation into Fxr1.

Further, the study also sufficiently proves Fxr1 is a putative substrate of Fbxo4 in HEK293T cells. First, it is superior to show molecular/genetic or pharmacological

intervention of Fbxo4 didn't alter Fxr1 gene expression but protein level. Second, multiple comprehensive biochemical assessments were conducted to show that SCFFbxo4 directly ubiquitylates and degrades Fxr1 that maybe in a manner that depends on GSK3 β -phosphorylated Fbxo4. Considering the previously established role of GSK3 β to phosphorylate Fxr1 for down-regulation, the study also used multiple firmly designed experiments to show co-expressing GSK3 β increase Fxr1 ubiquitylation in a SCFFbxo4 dependent manner, further detailing the molecular basis of such interaction. Considering the possibility that some unidentified components within SCFFbxo4 may also mediate Fxr1 recognition, the study also carefully used recombinant wild-type/mutant/ phospho-mimetic SCFFbxo4 protein to demonstrate a role of Fbxo4 phosphorylation in Fxr1 ubiquitylation.

In general, data from these sections are clear, sufficient, accurate, and compelling. However, considering GSK3 β targets multiple potent downstream factors that involve mechanisms such as supporting translation, which could be a mediator of Fxr1 protein level, it may not be sufficient to demonstrate "GSK3 β -dependent Fbxo4 phosphorylation" by simply overexpressing a sole wild type GSK3 β . Further experiments by using GSK3 β critical mutants may be needed to firmly prove this mechanism. In addition, it would be more compelling (may not be that critical) to conduct a few simple experiments to demonstrate whether all those genetic-interfering cells also showed attenuated translation efficiency. In this study, assessing of Fxr1 protein level is simply the final readout; however, it is the reduced protein level that may be affected not only by degradation, but may also be due to reduced translation efficiency.

Reply: We understand this concern and at this point we cannot rule out the possibility that GSK3 β may influence protein translation at some level. However, we do demonstrate a direct effect of GSK3 β on Fxr1 ubiquitylation. Furthermore, our previous data have demonstrated that GSK3 β directly phosphorylates Fbxo4 and this is required for its activity. An Fbxo4 mutant that cannot be phosphorylated by GSK3 β is not able to regulate Fxr1 ubiquitylation demonstrating a direct relationship between GSK3 β , Fbxo4 and Fxr1 ubiquitylation. In addition, the loss of function of phospho-dead Fbxo4 in driving down Fxr1 in HNSCC cells also strengthens our hypothesis - degradation instead of other mechanisms contributes to Fxr1 downregulation (Supplementary Fig. 7f-h). To directly answer the reviewer's question, we performed the ubiquitylation of Fxr1 using both WT and KD GSK3 β . Our new data strongly indicate GSK3 β phosphorylates and promotes the activation of Fbxo4, which leads to Fxr1 ubiquitylation (Supplementary Fig. 5f).

Figure 4:

Next, the study steps into investigating if such mechanism also exists in head neck cancer cell models. Data presented in this section is generally compelling but not sufficient. First, by simply looking at gene expression of Fxr1 in cancer cohorts may not be sufficient as this mechanism is completely established on protein level. It may be more compelling to move some data from Figure 6 into Figure 4 to further demonstrate Fxr1 protein is elevated, in general, in head and neck cancer tissue samples.

Reply: We totally agree with this. To clarify this and improve the presentation of our data, we moved Western blot data of frozen tissues from Previous Fig. 6 to Current (Fig. 3b).

Second, the paper didn't specifically state which HNSCC cell line is used for knockdown (Fbxo4) experiment and failed to present relevant data of knocking-down efficiency. If knocking down effect is not assessed by using w.t. Fbxo4-expressing cells, it would be more compelling to perform a rescue experiment by applying recombinant w.t. Fbxo4 protein into the knocking down experiment.

Reply: Actually, we knocked down Fbxo4 in all three HNSCC cell lines: 74A, 74B and SCC9 cells and we detected elevated Fxr1 protein levels (Fig. 3f). To address the reviewer's question, we overexpressed WT Fbxo4 in Fbxo4 knockdown cells (Fbxo4 shRNAs target 3'-UTR); as shown in (Supplementary Fig. 7d), WT Fbxo4 successfully antagonized Fxr1 upregulation-mediated by Fbxo4 knockdown.

Third, although it is not critical, it is worthy to note that changes of Fxr1 protein level in multiple data sets shown in Figure 4 are not compelling (considering the change of W.B band intensity), may bring a question that a compensation mechanism may raise due to genetic alteration in head and neck cancer cells. This needs to be discussed and addressed.

Reply: The reviewer's suggestion is very good. In order to improve the data presentation, we have put the band quantification data below Fxr1 bands (Refer to relative Figures). We have put a brief introduction of the compensation mechanism in the Discussion section.

Fourth, as such mechanism involves "GSK3 β -dependent Fbxo4 phosphorylation", it would be more compelling to show the "GSK3 β -dependent Fbxo4 phosphorylation" also exists in head and neck cancer cells. As GSK3 β is such an important cancer-associated factor that crosses with many critical signaling such as PI3K that regulates various cell functions including cell proliferation and senescence, it would be interesting to see whether interfering GSK3 β also affect Fxr1 protein level and relevant phenotypes. Many GSK3 β inhibitors are available. It only needs a simple experiment by treating such inhibitor in various already-established vector-expressing cells to assess Fxr1 protein level.

Reply: This is an excellent suggestion to make our model completed. Upon *in vivo* ubiquitylation assay using GSK3 β kinase-dead mutant, we have included the GSK3 β inhibitor (SB-216763) treatment to demonstrate the effects of GSK3 β inhibition on Fxr1 expression in all three HNSCC cell lines. As shown in (Supplementary Fig. 7e), GSK3 β inhibitor successfully rescued the Fbxo4-mediated Fxr1 downregulation.

Figures 5-7:

The study further investigated whether Fbxo4 regulates head and neck cancer cell proliferation and senescence. From the data (Fig 5. A/B/C), it seems that the effect of Fbxo4 knockdown on cellular proliferation only observed after day-3, bringing a question whether cells are confluent at day 3 and the difference only observed when cells reached

to confluency and the effect is more dependent on a cell-to-cell contact manner, rather than the putative cell growth. The paper didn't state/discuss this. It needs to be more clarified, especially in the context to assess senescence, which is highly dependent on cell confluency in an experimental perspective. Once again, the knockdown efficiency in relevant cells are not presented in the data set, which bring some smoke to assess the putative effect, especially multiple experiments are done by concurrent knocking down of both Fbxo4 and Fxr1 and in a scenario that knocking down of Fbxo4 itself enhances Fxr1 protein expression.

Reply: Good point, actually, we have the original plate pictures (cells from Day 5 before acetic acid extraction); however, due to space limitation, we didn't include these pictures for the first submission. We added them in the revised manuscript (Supplementary Fig. 8i-k). From these pictures, one can see that the cells are not confluent, ruling out the possibility of cell confluency-induced senescence. In fact, Fxr1 knockdown was performed in Fbxo4 knockdown cells. To make it clear, we have added the Western blot results to illustrate Fbxo4 was knocked down already, as showing in (Fig. 4b and Supplementary Fig. 8b,d).

In addition, for the β -Gal staining assay (Fig.5 G/H/I), it is also recommended to show a quantification data by assessing multiple bright-field records, not only showing a few representative images.

Reply: We have already added the quantification of the percentage of senescent cells based on three biological repeats.

In addition, it would be wise to also clarify the experimental conditions for these experiments, whether the experiments were done by using selection agent such as puromycin to force the knockdown efficiency as sometimes, selection agent would promote cellular senescence at different levels.

Reply: We agree with this point. Actually, we didn't select the cells with puromycin. Each cell line was transiently knocked-down of Fxr1 or overexpressed of Fbxo4 in order for senescent cell staining. To clarify this, we have already improved the Senescent Staining part in Materials and Methods.

Finally, the data (Fig.5 G/H/I) suggest that "Fbxo4 expression increased SA- β -Gal staining to a similar degree as Fxr1 knockdown", which brings some concerns that the reduced Fxr1 protein level is due to Fbxo4/GSK3 β -mediated degradation or simply due to Fbxo4-induced senescence, which translation could be halted. It is recommended to design and perform additional experiments such as polysome profiling to rule out the possibility that the reduced Fxr1 protein expression is due to the senescence.

Otherwise the study is disconnected between the molecular studies and the cancer cells – related studies, regardless correlation between Fbxo4 loss and Fxr1 overexpression in HNSCC tumors and relevant mouse models presented in Figure 6.

Reply: Good point; in fact, all our biochemical data highly supported the turnover of Fxr1 by Fbxo4 in HNSCC cells. To directly address the reviewer's concern, we tested Fxr1 turnover with Fbxo4 expression in HEK293T cells, which do not senesce in response to Fbxo4 overexpression. Our data clearly indicate the existence of

Fbxo4-mediated Fxr1 degradation independent of the senescence-induced translation inhibition (Supplementary Fig. 7c).

REVIEWERS' COMMENTS:

Reviewer #1 (Remarks to the Author):

The authors have addressed my concerns adequately.

Reviewer #2 (Remarks to the Author):

The revised version of the manuscript addressed my concerns regarding presentation, interpretation and deposition of computational modeling results. In this respect I believe that the paper is appropriate for publication

Minor comment: It would be great if in the final manuscript the authors provide models in PDB format with separate chain IDs for different proteins (currently it is single one chain PDB) - otherwise it is tricky to work with those structures.

Reviewer #3 (Remarks to the Author):

The authors have addressed most of my concerns and have strengthened the paper. In my opinion the manuscript is now worthy of publication in Nature Communications.

Reviewer #4 (Remarks to the Author):

The revised manuscript is well tailored; many efforts have been put into new experiments. Data from revised manuscript are complete.

1) Previously, it is concerned that the reduced Fxr1 protein level that may be affected not only by degradation, but may also be due to reduced translation efficiency as GSK3 β targets multiple potent downstream factors that involve translation regulation, which could be a mediator of Fxr1 protein level. Multiple experiments by using GSK3 β critical mutants were recommended. To answer this question, the author performed the ubiquitylation of Fxr1 using both WT and KD GSK3 β . The new data shown in Supplementary Fig. 5f (plus data from Supplementary Fig. 7f-h) cleared the concern.

2) By the previous advice, the author moved Western blot data of frozen tissues from Figure 6 into Figure 4 to make the flow more logical and compelling.

3) It is recommended to perform a rescue experiment by applying recombinant w.t. Fbxo4 protein into the knocking down experiment. Per advice, the author overexpressed WT Fbxo4 in Fbxo4 knockdown cells (Supplementary Fig. 7d) and successfully showed that WT Fbxo4 antagonized Fxr1 upregulation-mediated by Fbxo4 knockdown.

4) Considering the concern of weak band intensity in multiple data sets shown in Figure 4, the author quantified band intensity and nicely addressed and discussed compensation mechanism in the discussion, which make the whole data presentation more compelling.

5) Per recommendation, the author applied the GSK3 β inhibitor (SB-216763) treatment to demonstrate the effects of GSK3 β inhibition on Fxr1 expression in all three HNSCC cell lines (Supplementary Fig. 7e). GSK3 β inhibitor successfully rescued the Fbxo4-mediated Fxr1 downregulation. This experiment made the model completed.

6) Regarding the concern that the senescence effect may be due to cell confluency, the author added multiple data sets (Supplementary Fig. 8i-k), successfully ruling out this concern.

7) Regarding the concern that whether reduced Fxr1 protein level is due to Fbxo4/GSK3 β -mediated degradation or simply due to Fbxo4-induced senescence, which translation could be halted, the author tested Fxr1 turnover with Fbxo4 expression in HEK293T cells. The new data (Supplementary Fig. 7c) indicated that Fbxo4-mediated Fxr1 degradation is independent of the senescence-induced translation inhibition, which successfully cleared out this concern.

The revised manuscript thoroughly cleared multiple concerns from the previous manuscript. The new manuscript is convincing and complete. It is recommended to publish these findings.

Point-by-point Response to Referees

REVIEWERS' COMMENTS:

Reviewer #1 (Remarks to the Author):

The authors have addressed my concerns adequately.

Reviewer #2 (Remarks to the Author):

The revised version of the manuscript addressed my concerns regarding presentation, interpretation and deposition of computational modeling results. In this respect I believe that the paper is appropriate for publication.

Minor comment: It would be great if in the final manuscript the authors provide models in PDB format with separate chains IDs for different proteins (currently it is single one chain PDB) - otherwise it is tricky to work with those structures.

We have attached the corrected files as Supplementary Data 1-10 with Fbxo4 as Chain A and Fxr1 as Chain B for all ten pdb files. In addition, we added one statement in the Methods where we refer to the Supplementary Data that says, " Ten .pdb files are provided as Supplementary Data 1-10 named after the ClusPro rank and number of poses in the model cluster, in which Chain A is Fbxo4 and Chain B is Fxr1."

Reviewer #3 (Remarks to the Author):

The authors have addressed most of my concerns and have strengthened the paper. In my opinion the manuscript is now worthy of publication in Nature Communications.

Reviewer #4 (Remarks to the Author):

The revised manuscript is well tailored; many efforts have been put into new experiments. Data from revised manuscript are complete.

1) Previously, it is concerned that the reduced Fxr1 protein level that may be affected not only by degradation, but may also be due to reduced translation efficiency as GSK3 β targets multiple potent downstream factors that involve translation regulation, which could be a mediator of Fxr1 protein level. Multiple experiments by using GSK3 β critical mutants were recommended. To answer this question, the author performed the ubiquitylation of Fxr1 using both WT and KD GSK3 β . The new data shown in Supplementary Fig. 5f (plus data from Supplementary Fig. 7f-h) cleared the concern.

Thank you.

2) By the previous advice, the author moved Western blot data of frozen tissues from Figure 6 into Figure 4 to make the flow more logical and compelling.

Thank you.

3) It is recommended to perform a rescue experiment by applying recombinant w.t. Fbxo4 protein into the knocking down experiment. Per advice, the author overexpressed WT Fbxo4 in Fbxo4 knockdown cells (Supplementary Fig. 7d) and successfully showed that WT Fbxo4 antagonized Fxr1 upregulation-mediated by Fbxo4 knockdown.

Thank you.

4) Considering the concern of weak band intensity in multiple data sets shown in Figure 4, the author quantified band intensity and nicely addressed and discussed compensation mechanism in the discussion, which make the whole data presentation more compelling.

Thank you.

5) Per recommendation, the author applied the GSK3 β inhibitor (SB-216763) treatment to demonstrate the effects of GSK3 β inhibition on Fxr1 expression in all three HNSCC cell lines (Supplementary Fig. 7e). GSK3 β inhibitor successfully rescued the Fbxo4-mediated Fxr1 downregulation. This experiment made the model completed.

Thank you.

6) Regarding the concern that the senescence effect may due to cell confluency, the author added multiple data sets (Supplementary Fig. 8i-k), successfully ruled out this concern.

Thank you.

7) Regarding the concern that whether reduced Fxr1 protein level is due to Fbxo4/GSK3 β -mediated degradation or simply due to Fbxo4-induced senescence, which translation could be halted, the author tested Fxr1 turnover with Fbxo4 expression in HEK293T cells. The new data (Supplementary Fig. 7c) indicated that Fbxo4-mediated Fxr1 degradation independent of the senescence-induced translation inhibition, which successfully cleared out this concern.

Thank you.

The revised manuscript thoroughly cleared multiple concerns from previous manuscript. The new manuscript is convincing and complete. It is recommended to publish these findings.

Thank you.